# HyperMotion: DiT-Based Pose-Guided Human Image Animation of Complex Motions

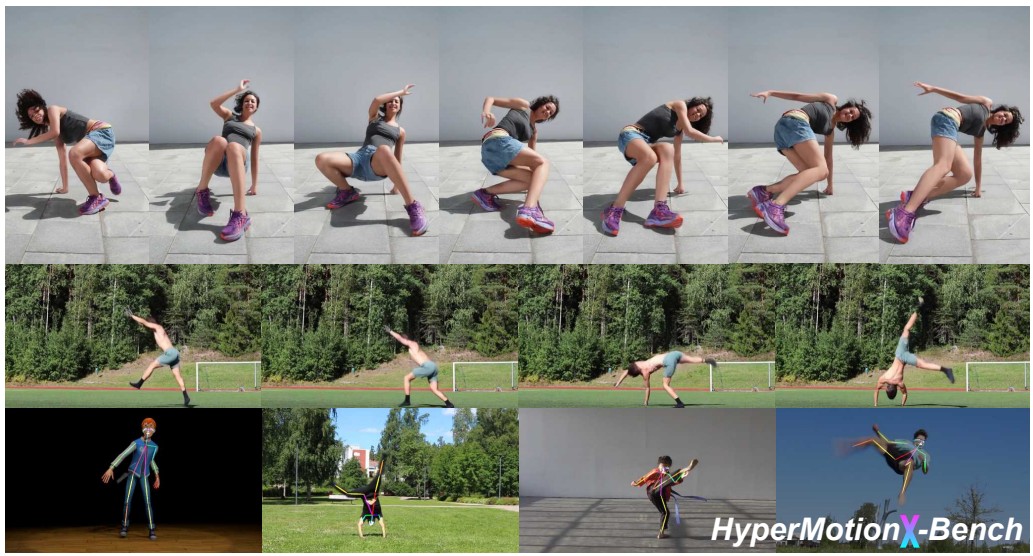

**Figure 1:** Complex human motion animation samples generated by our method. We present generation examples in both landscape (1024 × 576) and portrait (576 × 1024) resolutions.

## Abstract

Recent advances in diffusion models have significantly improved conditional video generation, particularly in the pose-guided human image animation task. Although existing methods are capable of generating high-fidelity and time-consistent animation sequences in regular motions and static scenes. However there are still obvious limitations when facing complex human body motions that contain highly dynamic, non-standard motions, and the lack of a high-quality benchmark for evaluation of complex human motion animations. To address this challenge, we propose a concise yet powerful DiT-based human animation generation baseline and design spatial low-frequency enhanced RoPE, a novel module that selectively enhances low-frequency spatial feature modeling by introducing learnable frequency scaling. Furthermore, we introduce the **Open-HyperMotionX Dataset** and **HyperMotionX Bench**, which provide high-quality human pose annotations and curated video clips for evaluating and improving pose-guided human image animation models under complex human motion conditions. Our method significantly improves structural stability and appearance consistency in highly dynamic human motion sequences. Extensive experiments demonstrate the effectiveness of our dataset and proposed approach in advancing the generation quality of complex human motion image animations. Codes will be made publicly available.

## 1 Introduction

With the rapid advancement of diffusion models, pose-guided human image animation (Hu, 2024; Moore Threads Corporation, 2024; Zhang et al., 2024; Zhu et al., 2024; Peng et al., 2024; Tan et al., 2024) have achieved remarkable progress. This task focuses on generating temporally coherent

human image sequences, conditioned on a reference image of the target person and a corresponding sequence of pose guidance. Recent approaches (Hu, 2024; Wang et al., 2025b; Jiang et al., 2023) that integrate human keypoint estimation, high-fidelity image reconstruction, and temporal modeling have demonstrated strong capabilities in synthesizing realistic animations under static scenes or routine actions. These techniques have been widely applied in virtual avatars, motion transfer, post-production, and other creative applications (Hu, 2024).

While existing methods (Hu, 2024; Zhang et al., 2024; Peng et al., 2024; Tan et al., 2024) have demonstrated promising performance in general cases, they still struggle to accurately reconstruct human image animation involving complex human motion dynamics, especially for **Hypermotion:** stunt actions, tricking motions, and acrobatic movements, defined as complex human actions with rapid and atypical motion dynamics. Existing methods typically rely on external human pose estimation methods such as DWpose (Yang et al., 2023), Openpose (Cao et al., 2019), RTMpose (Jiang et al., 2023) to extract driving pose sequences for animation generation. However, these approaches often fail to accurately capture pose sequences when applied to human videos involving complex motions, as illustrated in Fig. 2, resulting in a lack of high-quality pose guidance to generate realistic human animation in such scenarios. To address this gap, we propose a new **Open-HyperMotionX Dataset** and **HyperMotionX Bench**, centered around complex human motion video clips. The dataset is constructed using our proposed complex human motion region extraction pipeline and is enriched with high-quality human pose annotations for both model training and evaluation. Our first aim is to verify: *If we provide high-fidelity complex motion pose sequences, can existing human animation models successfully reconstruct human animations with complex motion dynamics?*

We have tested and observed that current mainstream approaches, even when accurate and high-quality pose sequences are provided as guidance, the generated image sequences often suffer from appearance and structural distortions, pose misalignment and pixel region disorder, which corresponds to the low-frequency features severe degradation of the human appearance in the video frames, rather than the texture, edge detail represented by the high frequencies. These issues become particularly prominent in unconventional motion frames (roll, flip, handspring,

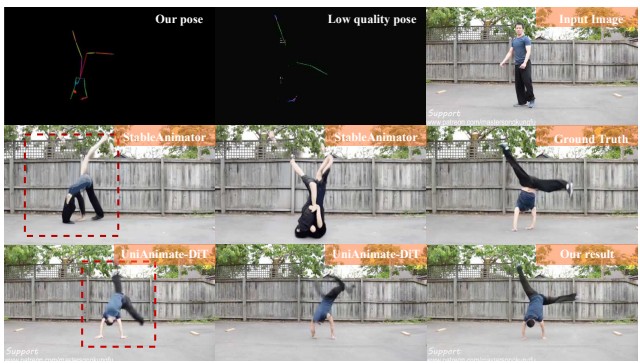

**Figure 2:** Sample video frames of previous methods. When we provide separate high-quality and low-quality pose.

round kick). The root cause lies in the insufficient spatial modeling capacity of current models for these non-standard, nonlinear action segments. These complex motion segments are also characterized by temporal sparsity and are present in only a small fraction of the total video frames. Thus, this temporal sparsity poses a challenge for the model to effectively learn the human body spatial representations in complex motion frames.

To address these issues, we argue that the core limitation stems from the insufficient modeling of low-frequency spatial features, which encode global structures and overall body appearance. Under the condition of having high-quality annotated pose sequences in the training dataset, enhancing the model's ability to capture low-frequency spatial features in complex human motion scenes becomes the key to further improving generation quality. To this end, we first propose a simple DiT-based baseline for the pose-guided human image animation task. It enables the generation of human image animation with large-scale subject motion in open scenes, conditioned on a given reference image and pose sequence. We then introduce Spatial Low-Frequency Enhancement Rotary Position Embedding (SLF-RoPE), a simple yet effective modification of the standard RoPE mechanism. SLF-RoPE selectively amplifies the lowest-frequency channels of the height and width within the frequency tensor by introducing two learnable scaling factors. This additional guidance improves the attention mechanism's ability to preserve human body structure, appearance, and spatial consistency, particularly under complex motion conditions, where conventional methods often suffer from structural degradation. Furthermore, considering the complex motion frames usually occupy a small portion of the video, we apply Wavelet transform to extract the optimal temporal window

through an energy-based optimization algorithm. Experiments and qualitative results have verified the effectiveness of our method in addressing the complex motion generation.

In summary, we make the following contributions:

- We propose a simple DiT-based baseline for the pose-guided human image animation task, capable of handling open-world scenarios and large-scale subject motions.

- We introduce Spatial Low-Frequency Enhanced Rotary Positional Embedding, a novel method to selectively enhance spatial low-frequency components, improving appearance and structural fidelity under complex human motions.

- We contribute Open-HyperMotionX Dataset and HyperMotionX Bench, providing high-quality human pose annotations and a benchmark to evaluate pose-guided human image animation models under complex human motion.

## 2 RELATED WORK

### 2.1 DIFFUSION FOR VIDEO GENERATION

Recent advances in video generation have been largely driven by diffusion-based frameworks, many of which are adapted from the Stable Diffusion architecture (Rombach et al., 2022). Early methods modify the UNet (Ronneberger et al., 2015) to incorporate temporal modeling. For example, VDM (Ho et al., 2022) extends 2D U-Net to 3D, while Animatediff (Guo et al., 2024) integrates 1D temporal attention into 2D spatial blocks for efficiency. More recently, transformer-based diffusion models such as DiT (Peebles & Xie, 2023) have shown superior performance in visual generation (Chen et al., 2023). These architectures have been adapted to video tasks (Hong et al., 2022), yielding variants that either use cross-attention for text embeddings (Peebles & Xie, 2023) or jointly attend to concatenated text and visual features (Esser et al., 2024). In terms of autoencoding, early models relied on standard VAEs (Kingma et al., 2013; Rombach et al., 2022), while recent works such as Hunyuan-Video (Kong et al., 2024) and Wan2.1 (Wang et al., 2025a) adopt 3D VAE architectures for improved compression and reconstruction. For textual conditioning, most recent methods employ the T5 family (Raffel et al., 2020) alongside CLIP (Radford et al., 2021). Additionally, Rotary Position Embedding (RoPE) (Su et al., 2024) has become a widely adopted technique for encoding positional information in diffusion transformer models.

### 2.2 POSE-GUIDED HUMAN IMAGE ANIMATION

Pose-guided human animation generation is the task of synthesizing temporally coherent, photo-realistic human videos whose appearance matches a given reference human image identity while following an input target pose sequence (Hu, 2024; Zhang et al., 2024). Early works (Li et al., 2019; Siarohin et al., 2019; 2021) based on Generative Adversarial Networks (GANs) (Goodfellow et al., 2020) to generate animation from source image. However, these GANs-based models often show various artifacts in the generated animation results. Recently, animation based on the diffusion models (Wang et al., 2024a) emerged and achieved pleasing human animation results. Hu (2024); Xu et al. (2024b) both include a pose net to process pose information and a reference net to model appearance, with the introduction of the temporal attention layer (Guo et al., 2024) enhancing temporal consistency. In addition, more representation information has been introduced such as depth and 3D signal SMPL to enhance the controllable capability, as represented by Champ (Zhu et al., 2024; Zhou et al., 2024). ControlNeXt (Peng et al., 2024) and Mimicmotion (Zhang et al., 2024) all utlize post-processing to deal with facial distortion. Stableanimator (Tu et al., 2024) introduced the face encoder and combined it with the face mask to address this problem. Animate-X (Tan et al., 2024) has developed the Pose Indicator to generalize the model to the animation of anthropomorphic characters. Follow-Your-Pose V2 (Xue et al., 2024) enhanced implicit decoupling towards multi-character image animation. UniAnimate (Wang et al., 2025b) introduced an additional Mamba module. HumanVid (Wang et al., 2024b) added the camera position parameter to the model, allowing the generation of animations that incorporate camera motion and provided a human image animation dataset. With the recent success of the video diffusion transformer (Peebles & Xie, 2023) model, UniAnimate-DiT (Wang et al., 2025b), DreamActor (Luo et al., 2025), have started to extend the task to a DiT-based model and have demonstrated high-quality generation performance,

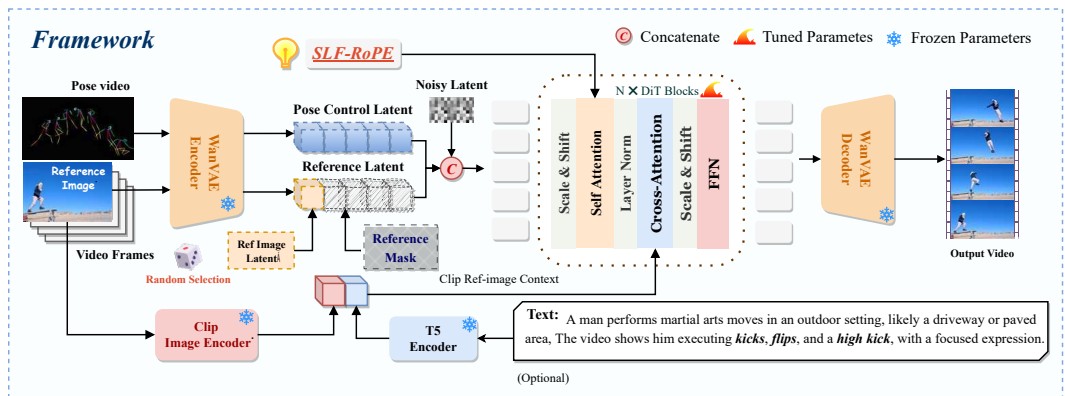

**Figure 3: The overview of our Hypermotion framework.** The model takes a reference image and a driving pose video as inputs and generates human animation. Pose control and reference image are injected via latent composition and guided by a binary mask. Spatial Low-Frequency Enhanced RoPE is applied in self-attention.

Despite promising results, current methods struggle on complex motions; even with high-quality pose inputs, outputs show structural artifacts and misalign with the driving poses.

## 3 METHODOLOGY

### 3.1 PRELIMINARIES

**Diffusion Transformer (DiT).** The DiT (Peebles & Xie, 2023) proposes a novel design that combines the generative strengths of diffusion models with the representational power of transformer architectures (Vaswani et al., 2017). This combination effectively addresses the inherent limitations of traditional UNet-based latent diffusion models (LDMs), resulting in improvements in generation quality, model flexibility, and scalability.

**Rotary Position Embedding (RoPE).** Rotary Positional Embedding (RoPE) (Su et al., 2024) has become the standard way of injecting position information into modern Transformer layers. For a 1-D token sequence, let $\boldsymbol{x} \in \mathbb{R}^d$ denote the input at position $p \in \mathbb{N}$. We select the first $d' \leq d$ channels (assumed even) and apply a position-dependent rotation to each channel pair:

$$\boldsymbol{f}^{\text{RoPE}}(\boldsymbol{x}, p, \boldsymbol{\theta})_j = \begin{bmatrix} \cos(p\,\theta_j) & -\sin(p\,\theta_j) \\ \sin(p\,\theta_j) & \cos(p\,\theta_j) \end{bmatrix} \begin{bmatrix} x_{2j} \\ x_{2j+1} \end{bmatrix}, \tag{1}$$

where $\boldsymbol{\theta} \in \mathbb{R}^{d'/2}$, with the frequency components defined as $\theta_j = b^{-2(j-1)/d'}$ for $j = 1, \ldots, d'/2$. The vector $\boldsymbol{\theta}$ specifies the positional encoding frequencies across all rotated channel dimensions, and $b$ serves as a scaling hyperparameter controlling the base frequency. The inner product between two RoPE-embedded vectors is dependent solely on their relative positional offset, enabling relative position encoding without introducing additional learnable parameters. In practical implementations, RoPE is applied to both query and key vectors prior to the dot-product operation within the attention mechanism, such that the resulting attention weights naturally encode relative positional information.

**Notations.** Let $\boldsymbol{L}_{\text{noisy}} \in \mathbb{R}^{B \times C \times T \times H \times W}$ denote the noisy latent video sequence, where $B$ is the batch size, $C$ is the channel dimension, $T$ is the frame length, and $H, W$ represent spatial height and width. The objective of Latent Video Diffusion Models (LVDM) is to progressively denoise $\boldsymbol{L}_{\text{noisy}}$ to produce realistic video samples in the latent space before decoding back to pixel space.

### 3.2 SIMPLE DiT-BASED BASELINE

Building upon the Wan2.1 (Wang et al., 2025a) and Wan2.1-Fun-Control I2V (Bubbliiiing, 2025) video generation models, we propose a simple DiT-based baseline for the pose-guided human image animation task with minimal architectural modifications, as illustrated in Fig. 3. Unlike prior

human image animation baselines, our method does not require additional reference networks (Hu, 2024), ControlNet (Zhang et al., 2023) modules, or insertion of reference tokens into the token sequence with corresponding adjustments to positional embeddings or Rotary Position Embedding (RoPE) (Su et al., 2023). Furthermore, it eliminates the need for a separately trained PoseNet (Hu, 2024). Despite its simplicity, the baseline can generate human animations across diverse, open-domain scenes by conditioning on a reference image and a driving pose sequence.

The Wan2.1 I2V baseline (Wang et al., 2025a) introduces a conditional image as the first frame and applies a Wan-VAE compression to obtain latent guidance features. The condition image $I \in \mathbb{R}^{C \times 1 \times H \times W}$ is concatenated with zero-filled frames to form $I_c$, then compressed to latent $z_c$. A binary mask $M$ marks the preserved frame for conditioning. Latent noise $z_t$, condition $z_c$, and the mask $m$ are concatenated and passed to the DiT backbone for generation. The reference image is encoded via CLIP (Radford et al., 2021), combined with text tokens from umT5 (Chung et al., 2023), and injected through the Decoupled cross-attention.

### 3.3 LATENT COMPOSITION FOR CONDITIONAL CONTROL

We used a unified latent space composition framework to inject both pose and reference image conditions into the DiT-based video generation pipeline.

**Pose Control Latent Inject.** To enable the pose sequence to serve as a condition for guiding animation, our baseline eliminates the need for an additional 3D convolution-based pose network requiring extra training. Instead, the input pose video $I_{\text{pose}}$ is directly processed by the Wan-VAE to obtain a pose control latent $\mathbf{L}_{\text{pose}} \in \mathbb{R}^{B \times 16 \times T \times H \times W}$ that matches the shape of the noise-added latent $\mathbf{L}_{\text{noisy}} \in \mathbb{R}^{B \times 16 \times T \times H \times W}$ features. The resulting pose control latent is then concatenated with the latents along the channel dimension $\mathcal{C}$ before being fed into the denoising network.

**Reference Image Inject.** The key challenge is how to inject the reference image such that any provided pose sequence can consistently drive the target appearance. The Wan2.1Wang et al. (2025a) I2V base model does not support this capability and often suffers from misalignment between the input image and the driving pose, resulting in identity inconsistency and frame artifacts. Thus, the reference image should not be concatenated directly with the noisy latent as in the original I2V baseline.

To address this, we propose a simple yet effective mechanism based on latent space composition and masking. If a reference image is provided, it is preprocessed and encoded by Wan-VAE to obtain a reference latent $\mathbf{L}_{\text{ref}} \in \mathbb{R}^{B \times 16 \times 1 \times H \times W}$. A zero-initialized tensor with the same shape as the noisy video latents $\mathbf{L}_{\text{noisy}} \in \mathbb{R}^{B \times 16 \times T \times H \times W}$ is created, and $\mathbf{L}_{\text{ref}}$ is inserted into the first frame (index 0). Meanwhile, a binary reference mask $\mathbf{M}_{\text{ref}} \in \{0, 1\}^{B \times 4 \times T \times H \times W}$ is constructed, where only the first frame is set to 1.0 and all others to 0. This mask explicitly controls the spatio-temporal influence of the reference image, ensuring that its appearance only guides the first frame and prevents leakage into subsequent frames. Finally, the reference latent tensor and the reference mask are concatenated with the pose control latents $\mathbf{L}_{\text{pose}}$ and the noisy video latent $\mathbf{L}_{\text{noisy}}$ along the channel dimension to form the final input for the denoising network:

$$L_{\text{final}} = \text{Concat}(\mathbf{L}_{\text{noisy}}, \ \mathbf{L}_{\text{pose}}, \ \mathbf{L}_{\text{ref}}, \ \mathbf{M}_{\text{ref}}) \in \mathbb{R}. \tag{2}$$

**Spatial Low-Frequency Enhanced RoPE (SLF-RoPE).** The SLF-RoPE is a concise yet effective modification to the frequency formulation of RoPE that selectively amplifies the lowest frequency channels of the height and width axes. Our intuition is that low-frequency spatial components encode large-scale shapes, structures, and basic appearance layouts, which are most critical for maintaining global consistency under fast motions.

Following the standard RoPE definition, the frequency vector $\boldsymbol{\theta} \in \mathbb{R}^{d'/2}$ is computed as:

$$\boldsymbol{\theta}_j = b^{-2(j-1)/d'}, \quad j = 1, \ldots, d'/2, \tag{3}$$

where $b$ is a base scaling factor. In SLF-RoPE, we partition $\boldsymbol{\theta}$ into temporal ($t$), height ($h$), and width ($w$) frequencies. For the spatial axes, we define the low-frequency range as the bottom $\alpha$ fraction of channels. An extra scaling factor $\gamma > 1$ is then applied to these low-frequency components:

$$\boldsymbol{\theta}_{\text{low}}^{(h)} = \gamma \cdot \boldsymbol{\theta}_{\text{low}}^{(h)}, \quad \boldsymbol{\theta}_{\text{low}}^{(w)} = \gamma \cdot \boldsymbol{\theta}_{\text{low}}^{(w)}. \tag{4}$$

In our design, $\alpha$ is set to 30%, and $\gamma$ is dynamically modulated by two learnable parameters: motion scale and space scale factor. The motion scale controls the scaling strength based on the estimated motion intensity of the input video, while the space scale factor controls the global amplification ratio of the spatial low-frequency bands. The modified frequency tensor is then applied identically as in standard RoPE during the complex-valued rotation in the attention mechanism. This dynamic enhancement allows self-attention layers to better preserve spatial layouts and structural consistency in highly dynamic motion regions. Full pseudocode is given in *Appendix 1*.

### 3.4 OPEN-HYPERMOTIONX DATASET

At present, there is no publicly available dataset specifically designed for the generation and evaluation of complex human motion videos with accurate pose sequence annotations. To address this gap, we propose a new dataset, the **Open-HyperMotionX**. We curate source videos from the open-source MotionX (Zhang et al., 2025; Lin et al., 2023) dataset, which was originally developed for tasks such as digital humans, motion capture, motion generation, and human mesh recovery. MotionX (Zhang et al., 2025; Lin et al., 2023) provides a large collection of human motion videos, including martial arts, gymnastics, stunt actions, and other highly dynamic performances, typically captured with a fixed camera and continuous long takes. However, the dataset has not been directly associated with video generation tasks before.

Through a dedicated data processing pipeline and additional annotations, we refine and extend the video and 2D keypoint annotations of MotionX (Zhang et al., 2025; Lin et al., 2023) to create Open-HyperMotionX, tailored for the study of complex motion human image animation.

The **Open-HyperMotionX** dataset contains 19,597 video clips comprising 2,507,811 frames, with a total duration of approximately 26 hours and 40 minutes. The average duration of a single clip is 5 seconds, with the majority ranging between 5 and 6 seconds. The dataset has an average frame rate of 26.4 fps, with most videos recorded at either 25 or 30 fps. In terms of spatial resolution, the dominant format is 1920×1080 or 1080×1920, which accounts for the majority of the clips. Further statistics and details are provided in Appendix *Section 7 8*.

**Data Process.**   For the pose-guided human image animation task, the ideal training data should consist of video clips with continuous shots (no cuts), full body visibility, minimal occlusion, and only slight camera movement. The source videos in MotionX (Lin et al., 2023) largely satisfy these criteria, which is a key reason why we chose to follow and build upon the MotionX (Lin et al., 2023) dataset. However, additional processing is required for video generation involving complex human motions. We performed the following processing steps:

- We applied YOLOv8 (Jocher et al., 2023) to keep clips with a single person; keypoint trajectories were linearly interpolated and outliers removed to denoise annotations.

- To target complex motion regions in long videos, we analyzed 2D keypoints with a **wavelet transform** and selected optimal windows via an energy-based criterion, capturing representative segments (e.g., the airborne phase of a cartwheel). Further method details in Appendix *Section A.1*.

- We reduced text region bias by detecting text with EasyOCR (AI, 2020) and applying Gaussian blur masks.

- Following EasyAnimate (Xu et al., 2024a), captions were first generated in batch by InternVL2.0 (Chen et al., 2024) and then refined with LLaMA-8B (AI@Meta, 2024), prioritizing action-related keywords while minimizing detail of scene appearance.

### 3.5 HYPERMOTIONX BENCH

Our initial motivation stems from the current limitation that accurate and high-quality pose sequences for complex human motions cannot be reliably extracted using existing models such as DWpose (Yang et al., 2023), Openpose (Cao et al., 2019), RTMpose (Jiang et al., 2023). This raises the question: *if more accurate and refined pose sequences were provided to existing methods, could they successfully generate high-quality videos of complex human motions?* Furthermore, due to the lack of a dedicated benchmark for evaluating complex motion pose-guided human image video generation, we introduce the **HypermotionX Bench** as an additional dataset for our primary task of pose-driven human image animation.

**HypermotionX Bench** is designed to assess the quality of complex human motion generation while providing more accurate pose annotations and the corresponding pose-guided video sequences. The source videos were manually collected and selected from publicly available websites that meet open source criteria and processed following the same pipeline as Open-HyperMotionX. After testing, we found that Xpose (Yang et al., 2024) extracts the best performance of pose in complex human motion videos, so we use Xpose (Yang et al., 2024) to extract the whole body pose for our bench, due to the low accuracy of hand detection, we discarded the hand annotation for some of the most complex cases frames. After careful manual selection, the final benchmark contains 100 complex human motion video clips, primarily in landscape orientation with resolutions centered around 1080p. The benchmark covers a wide range of complex and basic human actions, with the complex motions mainly sourced from Tricking (martial arts), including classic movements such as Forward Rolls, Backward Rolls, Front Handspring, Back Handspring, Cartwheel, Front Flips, Webster, Back Flips, Moon Kick, Side Flips, and 360 Kick. Finally, it is important to note that, to ensure fair evaluation, all cases included in the HypermotionX Benchmark were strictly separated from the Open-HyperMotionX dataset used for training. Further HyperMotionX Benchmark visualizations are provided in *Appendix 6*.

## 4 EXPERIMENTS

### 4.1 IMPLEMENTATION DETAILS

Based on the Wan2.1-Fun-Control (Bubbliiiing, 2025) pre-training weights, we conducted training and experiments on the 14B parameters version of the model using our Open-HyperMotionX dataset. We conducted extensive experiments based on the Wan2.1-Fun framework, trained on our proposed Open-HyperMotionX dataset. For the 14B parameter model, we performed full fine-tuning for a total of 30,000 steps using the ZeRO Stage 2 (Rajbhandari et al., 2020) optimization strategy on 8 NVIDIA H20 GPUs, which required approximately 12 days of training. The learning rate for experiments was uniformly set to $5 \times 10^{-6}$. During inference, we set the Guide scale to 6.0, only a single RTX 3090 GPU is required which consistently produced satisfactory generation quality in various test samples.

In implementing **SLF-RoPE**, we replaced the original Rotary Positional Embedding (RoPE) used in the self-attention modules and introduced two learnable scaling parameters: the Motion Scale and the Space scale factor. The motion scale was initialized to 1.5 and the space scale factor was initialized to 0.02. Additionally, within the **SLF-RoPE** module, we set the low ratio to 30%, which determines the proportion of low-frequency channels along the spatial height and width dimensions.

**Table 1:** Quantitative comparison with state-of-the-art methods on the HyperMotionX Bench. ↑ indicates higher is better; ↓ indicates lower is better. Red and Blue denote the best and second-best, respectively.

| Method | PSNR ↑ | SSIM ↑ | L1 ↓ | LPIPS ↓ | FID ↓ | VFID ↓ | FVD ↓ | PCK@0.5(%) ↑ |
|--------|--------|--------|------|---------|-------|--------|-------|--------------|
| Animate-X (Tan et al., 2024) | 19.19 | 0.62 | $61.0 \times 10^{-3}$ | 0.210 | 110.32 | 17.07 | 1798.89 | 40.56 |
| ControlNeXt (Peng et al., 2024) | 20.39 | 0.64 | $58.9 \times 10^{-3}$ | 0.158 | 94.66 | 16.62 | 985.00 | 49.58 |
| StableAnimator (Tu et al., 2024) | 19.82 | 0.63 | $62.9 \times 10^{-3}$ | 0.163 | 93.22 | 33.13 | 1184.88 | 17.84 |
| UniAnimate-DiT (Wang et al., 2025b) | 20.90 | 0.68 | $56.9 \times 10^{-3}$ | 0.152 | 80.90 | 14.50 | 981.43 | 23.17 |
| Wan2.1-Fun-V1.1 (Bubbliiiing, 2025) | 20.72 | 0.63 | $58.0 \times 10^{-3}$ | 0.148 | 81.86 | 17.89 | 801.74 | 65.62 |
| **Hypermotion** | 22.03 | 0.71 | $47.0 \times 10^{-3}$ | 0.124 | 80.91 | 16.49 | 825.68 | 70.32 |

### 4.2 EXPERIMENTAL SETUP

As our work focuses on complex human animation and we propose the HypermotionX Bench specifically to evaluate this task, we provide not only the dataset, but also high-quality pose sequences and a reproducible method to extract these poses. To ensure fairness in evaluation, we made every effort to allow all comparison methods to use the same pose annotation provided by our benchmark.

Since there is no unified standard for frame rate, frame count, or resolution across different baseline methods, we carefully adapted the input settings of each method to be compatible with the HypermotionX Bench while preserving the integrity of a fair comparison. Specifically, for ControlNeXt (Peng et al., 2024; Tu et al., 2024), we used the pre-processed pose videos provided by us

**Figure 4:** Qualitative comparison between our method and previous state-of-the-art methods.Our method demonstrates superior structural coherence, appearance consistency, and motion stability under complex human motion such as front flip. Results are shown in both 1024×576 (landscape) and 576×1024 (portrait) resolutions.

and adjusted the resolution, frame rate, and frame count accordingly. Similarly, for UniAnimate-DiT (Wang et al., 2025b), the resolution was set to match our preprocessed pose videos; however, due to its fixed frame count requirement, we dynamically adjusted the frame rate to align with the duration of the pose video provided. For Animate-X (Tan et al., 2024), since the frame rate is fixed at 8 fps, the total frame number was calculated directly based on the duration of the corresponding pose video.

**Table 2:** Ablation study of the effectiveness of the proposed Open-HyperMotion dataset. We utilize the metrics of VBench-I2V (Huang et al., 2024) for evaluation.

| Method | Background Consistency ↑ | Overall Consistency ↑ | Motion Smooth ↑ |
|---|---|---|---|
| Wan2.1-Fun-Control I2V (Un-trained) | 94.14 | 10.57 | 98.99 |
| Wan2.1-Fun-Control I2V (Trained) | 94.61 | 10.61 | 99.08 |
| **Hypermotion**(Ours) | 95.00 | 10.54 | 99.19 |

We conducted a comprehensive quantitative evaluation of our method and recent state-of-the-art baselines on the proposed HyperMotionX Bench. Additionally, we evaluated the effectiveness of our dataset for training pose-driven human image animation models. We adopted widely used metrics including Pixel fidelity: PSNR (Hore & Ziou, 2010), SSIM (Wang et al., 2004), L1-Loss and Perceptual quality: FID (Heusel et al., 2017), LPIPS (Zhang et al., 2018), VFID (Balaji et al., 2019), FVD (Unterthiner et al., 2018). These metrics jointly assess both pixel-level reconstruction quality and perceptual-level fidelity, and are critical for evaluating the visual quality of human body generation in our task. Notably, we have introduced an additional quantitative metric in the benchmark: Percentage of Correct Keypoints (PCK) (Bergman et al., 2023). This metric calculates the percentage of 2D keypoints detected on a generated frame that fall within a specified error threshold relative to the ground truth (conditioning) keypoints. It provides a more direct evaluation of pose structural fidelity in the generated results.

In addition, to evaluate the value of our data set and the validity of our approach in our method, we introduce the three dimensions: Background consistency, overall consistency, smooth motion of Vbench-I2V (Huang et al., 2024) for the quantitative evaluation of no real video participation with our base model Wan2.1-FunV1.1-Control (Bubbliiiing, 2025) I2V.

## 4.3 QUANTITATIVE RESULTS

We compare our method with recent state-of-the-art baselines Animate-X (Tan et al., 2024), ControlNeX (Peng et al., 2024), StableAnimator (Tu et al., 2024), UniAnimate-DiT (Wang et al., 2025b), Wan2.1-Fun-V1.1 (Bubbliiiing, 2025), on the HyperMotionX Bench. Uniformly use the pose video provided in our bench as the input condition. As shown in Table 1, our model achieves either the best or second-best performance across all metrics. Notably, our method delivers significant gains on pixel-level fidelity metrics (PSNR, SSIM, L1). Moreover, on structural accuracy, measured by PCK, our approach also achieves SOTA on HyperMotionX bench with a substantial margin over all baselines. These results indicate that the proposed **SLF-RoPE** effectively enhances spatial structure while preserving low-frequency appearance in complex-motion frames. On perceptual and temporal metrics (LPIPS, FID, VFID, FVD) our method remains highly competitive and consistently ranks among the top performers. Taken together, the strong performance across low-level and high-level metrics demonstrates the effectiveness of our design for challenging human motion scenarios.

To further validate, We additionally adopt **VBench-I2V** as a no-reference evaluation protocol. We compare with our base model Wan2.1-Fun-V1.1-Control (I2V mode). As shown in Table 2, our method achieves superior performance across all three evaluation metrics: *Background Consistency*, *Overall Consistency*, and *Motion Smoothness*. These results further demonstrate the effectiveness of both our modeling framework and the proposed dataset in handling complex motion generation scenarios. In Fig. 4, we present qualitative comparisons with recent state-of-the-art baselines on complex human motion scenarios from the HyperMotionX Bench. For additional qualitative results in Appendix *Section* **9, 10**.

## 4.4 ABLATION STUDY

Ablation on **SLF-RoPE**. To analyse the contribution of SLF-RoPE we removed SLF-RoPE from our baseline and compared it to our baseline with the same number of training steps. As shown in Table 3, SLF-RoPE brings significant improvements to the model. For the Hypermotion model, SLF-RoPE contributes to improvements on pixel fidelity metrics (PSNR, SSIM, L1). On perceptual metrics (LPIPS, FID, VFID, FVD), we also observe improvements, albeit more modest than those over the baseline trained solely on our dataset. The slight decrease in perceptual performance is due to the perception–distortion (perceptual–fidelity) trade-off, rather than ineffectiveness of SLF-RoPE (Blau & Michaeli, 2017; Wang et al., 2025c). By contrast, SLF-RoPE delivers a clear gain in structural alignment, as evidenced by higher PCK score. Visual ablation results Appendix *Section* **8**.

**Table 3:** Ablation study on the impact of SLF-RoPE and proposed Open-HyperMotion dataset on the HyperMotionX Bench. ↑ indicates higher is better; ↓ indicates lower is better.

| Method | PSNR ↑ | SSIM ↑ | L1 ↓ | LPIPS ↓ | FID ↓ | VFID ↓ | FVD ↓ | PCK@0.5(%) ↑ |
|---|---|---|---|---|---|---|---|---|
| Baseline (Un-trained) | 20.72 | 0.63 | $58.0 \times 10^{-3}$ | 0.148 | 81.86 | 17.89 | 801.73 | 65.62 |
| Baseline (Trained) | 21.78 | 0.69 | $48.3 \times 10^{-3}$ | 0.125 | 77.00 | 13.19 | 758.71 | 68.10 |
| SLF-RoPE | 22.03 | 0.71 | $47.0 \times 10^{-3}$ | 0.124 | 80.91 | 16.49 | 825.68 | 70.32 |

We further evaluate the impact of SLF-RoPE on general-motion benchmarks, including TikTok (Jafarian & Park, 2021) and UBC-Fashion (Zablotskaia et al., 2019). As shown in the Table **9**, the results indicate that SLF-RoPE does not degrade the performance in these common motion datasets, achieving fidelity and perceptual quality comparable to baseline. This confirms that our frequency aware modification remains stable on simple motions while primarily benefiting complex motion scenarios.

In addition, we further analyze the impact of training on the Open-HyperMotionX dataset by evaluating the EasyAnimate5-7B Xu et al. (2024a) model (control-task) before and after finetuning, using the HyperMotionX Benchmark as a testbed. As shown in Table 4, finetuning leads to consistent improvements in most metrics, including PSNR, SSIM, L1, LPIPS, FID, and FVD, indicating enhanced visual quality, temporal coherence, and pixel-level fidelity. This shows that the proposed Open-HyperMotionX dataset effectively increases the ability of a different baselines to model complex human motions animation.

**Table 4:** Comparison of EasyAnimate5-7B Xu et al. (2024a) (control-task) before and after training on Open-HyperMotionX, evaluated on the HyperMotionX Benchmark. ↑ indicates higher is better; ↓ indicates lower is better.

| Method | PSNR ↑ | SSIM ↑ | L1 ↓ | LPIPS ↓ | FID ↓ | FVD ↓ |
|---|---|---|---|---|---|---|
| Un-trained | 19.31 | 0.64 | $63.9 \times 10^{-3}$ | 0.196 | 111.09 | 1377.32 |
| Trained | 19.74 | 0.66 | $59.4 \times 10^{-3}$ | 0.188 | 97.44 | 1129.42 |

**Table 5:** SLF-RoPE ablation study under low-quality pose (DWpose Yang et al. (2023)) inputs on the HyperMotionX Bench. ↑ indicates higher is better; ↓ indicates lower is better.

| Method | PSNR ↑ | SSIM ↑ | L1 ↓ | LPIPS ↓ | FID ↓ | FVD ↓ |
|---|---|---|---|---|---|---|
| Remove SLF-RoPE | 20.65 | 0.63 | $60.3 \times 10^{-3}$ | 0.1523 | 81.04 | 833.58 |
| SLF-RoPE | 21.26 | 0.67 | $52.1 \times 10^{-3}$ | 0.1366 | 83.031 | 882.51 |

The effect of SLF-RoPE under Low Quality Pose, as show in Table 5. To assess the robustness of SLF-RoPE, we replace high-quality pose inputs with degraded poses generated by DWpose Yang et al. (2023). Table 5 shows that incorporating SLF-RoPE still improves PSNR, SSIM, and L1, demonstrating that the module remains effective even when pose guidance is partially missing or noisy. This aligns with our motivation: SLF-RoPE leverages the diffusion prior to compensate for unreliable structural input, yielding more stable and coherent results.

**Table 6:** SLF-RoPE Ablation Study Using Different Frequency Domain Enhancement Strategies Training on HyperMotionX Bench. ↑ indicates higher is better; ↓ indicates lower is better.

| Model | PSNR ↑ | SSIM ↑ | L1 ↓ ($\times 10^{-3}$) | LPIPS ↓ | FID ↓ | FVD ↓ |
|---|---|---|---|---|---|---|
| No Enhance | 21.78 | 0.69 | 48.3 | 0.1255 | 77.00 | 758.71 |
| Enhance H | 21.14 | 0.68 | 52.4 | 0.1409 | 85.68 | 827.27 |
| Enhance L | 22.03 | 0.71 | 47.0 | 0.124 | 80.91 | 825.68 |

Finally, we compare different variants of frequency-domain enhancement within SLF-RoPE. As shown in Table 6, enhancing the low-frequency band (Enhance L) yields the best overall performance across reconstruction and perceptual metrics, confirming that low-frequency positional stability plays a dominant role in preserving large-scale spatial structure under complex human motion. In contrast, enhancing high-frequency signals (Enhance H) leads to degradation, further validating our design choice. It should be noted that during training, we employed identical training steps and hyperparameter settings.

## 5 CONCLUSION

In this work, we propose a simple yet effective DiT-based baseline and introduced the **SLF-RoPE** module to enhance spatial low-frequency modeling for pose-guided human image animation under complex motion conditions. We presented the Open-HyperMotionX Dataset and HyperMotionX Bench to establish a new benchmark for evaluating this challenging task. Extensive experiments on these benchmarks demonstrate that our method consistently outperforms existing SOTA approaches across both pixel-level metrics (PSNR, SSIM, L1), structural metric (PCK) and perceptual metrics (LPIPS, FVD, FID, VFID). Notably, gains in pixel-level accuracy and structural alignment validate that our proposed SLF-RoPE better enhances global appearance and structure. Qualitative results further confirm superior consistency of identity, accuracy of the limb, and motion stability in various scenarios. We hope our dataset, benchmark, and method will benefit the research community.

ETHICS STATEMENT

This work relies on publicly available datasets under their respective licenses. No new data involving human subjects were collected. All visualizations respect privacy. We confirm that our method and experiments do not raise additional ethical concerns

REPRODUCIBILITY STATEMENT

After the blind review period, we will release our codebase, training/inference scripts, configuration files, and model checkpoints, together with step-by-step instructions and evaluation protocols to fully reproduce all tables and figures

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

# A APPENDIX

CONTENTS

## A.1 COMPLEX HUMAN MOTIONS VIDEO CLIP EXTRACTION VIA WAVELET TRANSFORMER

As mentioned in Section 3.4 Data Process, To extract the most representative action segments from long video sequences, we developed an automated method based on human keypoint motion characteristics. This method utilizes the wavelet transform to analyze human keypoint movement patterns and employs an energy optimization algorithm to select the optimal time window. The specific steps are as follows:

### A.1.1 MOTION VELOCITY CALCULATION

Based on the extracted keypoint sequences, we calculate the inter-frame velocity of a specific joint (default index 0, typically corresponding to the head or neck):

$$v(t) = \sqrt{(x_{t+1} - x_t)^2 + (y_{t+1} - y_t)^2} \times \text{fps}. \tag{5}$$

where $x_t$ and $y_t$ are the coordinates of the joint at frame $t$, and $fps$ is the frame rate of the video. This step converts spatial position changes into time series data, providing a foundation for subsequent analysis.

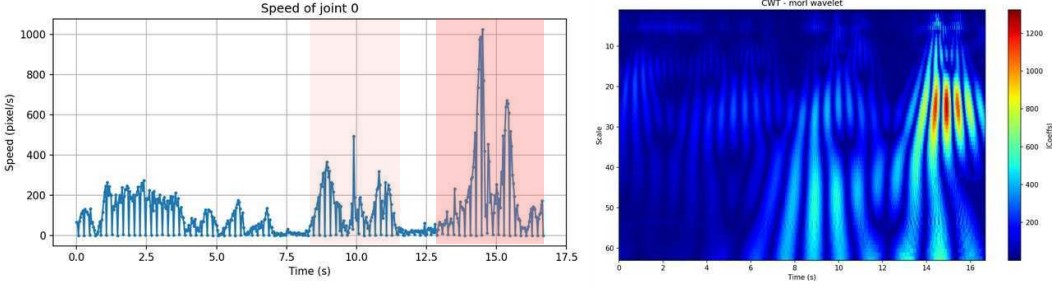

**Figure 5:** To extract the most representative action segments from long video sequences, we developed an automated method based on human keypoint motion characteristics. This method utilizes wavelet transform to analyze human keypoint movement patterns and employs an energy optimization algorithm to select the optimal time window. This figure shows the complex motion periods we extracted, corresponding to the vacating periods during the flip.

### A.1.2 WAVELET TRANSFORM ENERGY ANALYSIS

To capture time-frequency characteristics in the motion, we apply **Continuous Wavelet Transform (CWT)** to the velocity sequence:

$$CWT(a,b) = \frac{1}{\sqrt{a}} \int_{-\infty}^{\infty} v(t)\psi^*(\frac{t-b}{a})dt, \tag{6}$$

where $\psi$ is the Morlet wavelet function, $a$ is the scale parameter, and $b$ is the translation parameter. We calculate wavelet coefficients across multiple scales (1 to 128) and obtain an energy sequence by taking the absolute value of all scale coefficients and summing them:

$$E(t) = \sum_{a=1}^{a_{max}} |CWT(a,t)|. \tag{7}$$

This energy sequence reflects the complexity and intensity of motion at different time points.

### A.1.3 ENERGY PEAK OPTIMIZATION

To enhance the robustness of the algorithm, we apply peak width filtering to remove narrow peaks that may be caused by noise (the noise in keypoint velocity curves is primarily introduced by sporadic errors in keypoint localization, often leading to sharp, nonphysical jumps between adjacent frames.):

$$E'(t) = \begin{cases} E(t) & \text{if peak width} \geq \text{threshold} \\ 0 & \text{otherwise} \end{cases}. \tag{8}$$

In our implementation, the peak width threshold is set to 3 frames to ensure that only meaningful movements with sufficient duration are retained.

### A.1.4 OPTIMAL TIME WINDOW SELECTION

Finally, we employ a sliding window approach to find the best segment of fixed length (default 6 seconds) on the processed energy sequence:

$$W_{best} = \arg\max_i \sum_{j=i}^{i+L-1} E'(j), \tag{9}$$

where $L$ is the target window length (in frames). To handle boundary cases, when the optimal window is too close to the beginning or end of the video (less than 10 frames away), we adjust the window to start from the beginning or end 6 seconds before the end of the video. For videos shorter than the target length, we retain the original video without cropping. During the early stage of constructing Open-HyperMotionX, We identified numerous cases of missing annotations and noisy data from the original MotionX dataset (Lin et al., 2023) then manually screened the original data. The automated curation pipeline we proposed was designed precisely in response to these initial observations.

### A.1.5 VIDEO SEGMENT EXTRACTION IMPLEMENTATION

Based on the above method, we use FFmpeg for precise video cropping, ensuring that the output segments have consistent duration and contain the most representative actions. To ensure video quality, we re-encode the video segments using H.264 encoding (CRF=23).

Compared to traditional fixed time point sampling or manual selection, this method can automatically identify and extract video segments containing the richest motion information, providing more consistent and high-quality input data for subsequent action recognition and analysis.

### A.2 OPEN-HYPERMOTIONX DATASET STATISTICS AND DETAILS

We have provided a detailed comparison table of statistical metrics between our collected dataset and existing publically available datasets: TikTok dataset (Jafarian & Park, 2021), UBC-fashion (Zablotskaia et al., 2019), HumanVid (Wang et al., 2024b)(As many of the official links provided are no

longer available, we were only able to download 10,136 videos.) and our base dataset motionX (Lin et al., 2023). The Table 7 includes clip counts, average durations, resolutions, these statistics clearly highlight the advantages of our dataset over existing datasets in terms of data volume, resolution, video specification and duration distribution.

**Table 7:** Comparison of our Open-HyperMotion dataset with existing datasets.

| Dataset | Clips | Total Frames | Avg.Frames | Avg.Duration | Resolution | Total Duration | FPS |
|---|---|---|---|---|---|---|---|
| TikTok-dataset | 340 | 92,961 | 273 | 9s | 604×1080 | 3,098s 51min | 30 |
| UBC-fashion | 600 | 231,373 | 385 | 12.8s | 720×940 | 7,719s 128min | 30 |
| HumanVid | 10136 | 4,533,721 | 447 | 17.6s | 1080×1920 1440×2732 | 7,719s 128min | 25/30 |
| MotionX | 25,859 | 4,428,694 | 171 | 7s | 1920×1080 1280×720 852×480 640×360 | 163,567s 2,726min | 25/30 |
| **HyperMotionX** | **19,597** | **2,507,811** | **128** | **5.2s** | **1920×1080 1280×720** | **96,040s 1,600min** | **25/30** |

**Table 8:** Comparison of our Open-HyperMotion dataset with existing datasets about Motion complexity metrics. ↑ indicates higher values in Local Motion Mean and Global Displacement indicate greater motion complexity.

| Dataset | Local Motion Mean ↑ | Local Motion Std | Global Displacement Mean ↑ | Global Disp. Std |
|---|---|---|---|---|
| TikTok | $1.5 \times 10^{-2}$ | $8.6 \times 10^{-3}$ | $8.8 \times 10^{-3}$ | $8.4 \times 10^{-3}$ |
| UBCFashion | $6.0 \times 10^{-3}$ | $2.0 \times 10^{-3}$ | $5.7 \times 10^{-3}$ | $2.8 \times 10^{-3}$ |
| HumanVid | $1.1 \times 10^{-2}$ | $1.8 \times 10^{-2}$ | $\mathbf{4.7 \times 10^{-2}}$ | $7.8 \times 10^{-1}$ |
| MotionX | $6.5 \times 10^{-3}$ | $6.7 \times 10^{-3}$ | $1.1 \times 10^{-2}$ | $8.5 \times 10^{-1}$ |
| **HypermotionX** | $\mathbf{2.3 \times 10^{-2}}$ | $\mathbf{1.4 \times 10^{-2}}$ | $\mathbf{3.0 \times 10^{-2}}$ | $\mathbf{2.9 \times 10^{-2}}$ |

**Local Motion Mean** measures the average pixels movement of individual key points between frames, reflecting the intensity of fine-grained movements of the body parts. **Global Displacement Mean** quantifies the average frame-to-frame positional shift of the entire body, reflecting large-scale motion across the video. The local motion mean is defined as

$$\text{LMM} = \frac{1}{(T-1)K} \sum_{t=1}^{T-1} \sum_{k=1}^{K} \|\mathbf{p}_{t+1,k} - \mathbf{p}_{t,k}\|_2, \tag{10}$$

where $\mathbf{p}_{t,k} \in \mathbb{R}^2$ denotes the pixel coordinates of keypoint $k$ at frame $t$, $K$ is the number of keypoints, and $T$ is the number of frames.

The **global displacement mean** metric is calculated as the average Euclidean distance between the center points of the subject (one keypoint, body points in COCO format) across consecutive frames, normalized to frame size. The global displacement mean is defined as

$$\text{GDM} = \frac{1}{T-1} \sum_{t=1}^{T-1} \|\mathbf{c}_{t+1} - \mathbf{c}_t\|_2, \tag{11}$$

where $\mathbf{c}_t$ denotes the global position in frame $t$ and $T$ is the number of frames. It reflects the global movement of the body (e.g. jumping, running) and is not a weighted sum of local joint motion. It complements local motion metrics by capturing spatial displacement rather than articulation alone.

We introduce this metric to highlight a key difference between our dataset and others like TikTok dataset (Jafarian & Park, 2021) or UBCFashion (Zablotskaia et al., 2019) dataset, where subjects typically stay in place and perform on-the-spot motions. In contrast, our dataset features large-range global motion across the scene, from one side of the screen to the other. Although the values

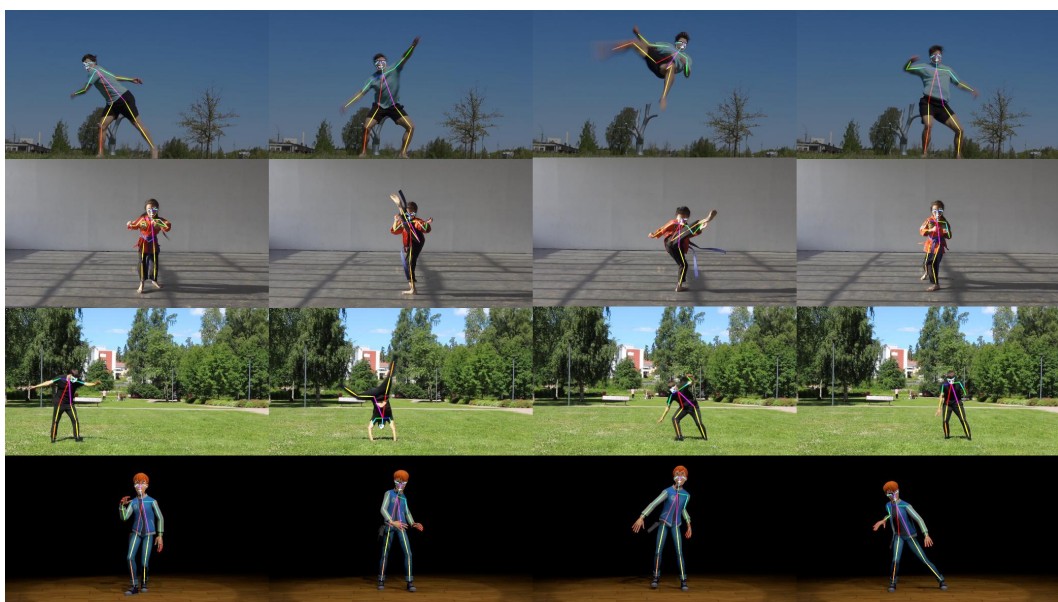

**Figure 6: Examples from our HyperMotionX Bench.** We contribute high-quality pose sequence annotations that contain a diversity of complex motion videos as well as different types of characters, including adults, children, and different styles of videos covering both real and cartoon scenes.

appear small, they are in normalized units. For a 512×512 video, this corresponds to approximately 20 pixels of movement per frame, which accumulates to hundreds of pixels over a sequence — indicating significant real-world movement.

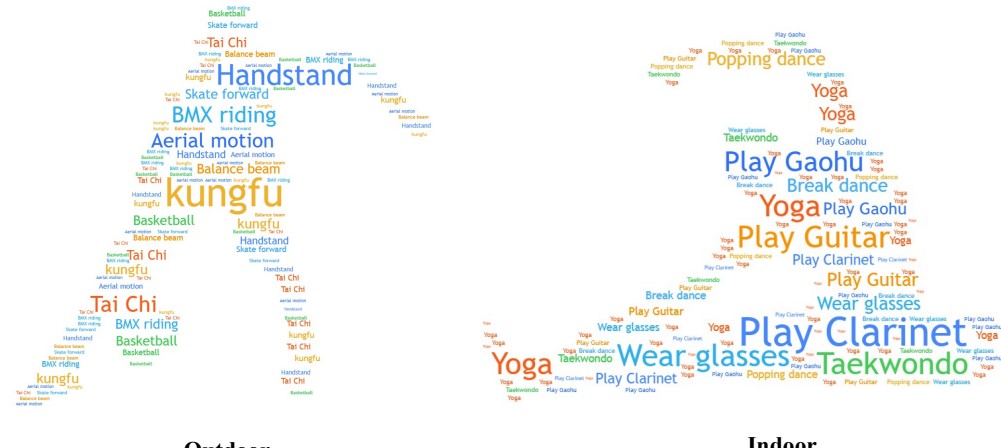

**Outdoor**  **Indoor**

**Figure 7: Motion categories in Open-HyperMotionX** The dataset encompasses diverse indoor and outdoor human motion categories of varying complexity.

In addition, for the **HyperMotionX Benchmark**, all 100 videos were manually selected by human annotators. We also found that most existing pose extraction methods including the current state-of-the-art XPose struggle to accurately capture hand keypoints under high-speed and complex motion. We also found that most existing pose extraction methods including current state-of-the-art XPose struggle to accurately capture hand keypoints under high-speed and complex motion. These manual interventions were essential steps we took to ensure the data quality of both training and evaluation sets.

As shown in Fig 6 **HyperMotionX Bench** also includes cartoon-style videos, ensuring diversity in visual style. The benchmark features subjects of varying genders and age groups, including children, and reflects balanced skin tones and ethnic diversity to support fair and inclusive evaluation.

Our 100 manually curated videos were selected from publicly available, legally safe internet sources. We believe this is a sufficient and representative set for assessing human motion quality in generated videos. We hope that this bench can be used in the community as a measure of the challenge of generating complex human motion with video generation models!

## A.3 EXAMPLES FROM ABLATION STUDY RESULT

Visualization of the ablation study in SLF-RoPE on the HyperMotionX Bench, as shown in Fig 6. In the first and second cases, we highlight the regions with red zoom-in boxes for detailed comparison. We observe that removing SLF-RoPE and relying solely on our baseline leads to inconsistencies between the generated human appearance and the input reference image. In the first case, the person in the reference image is wearing long black trousers, but the baseline model without SLF-RoPE instead generates short black pants. The discrepancy is even more pronounced in the second case, where the model fails to accurately reproduce the black T-shirt from the reference image.

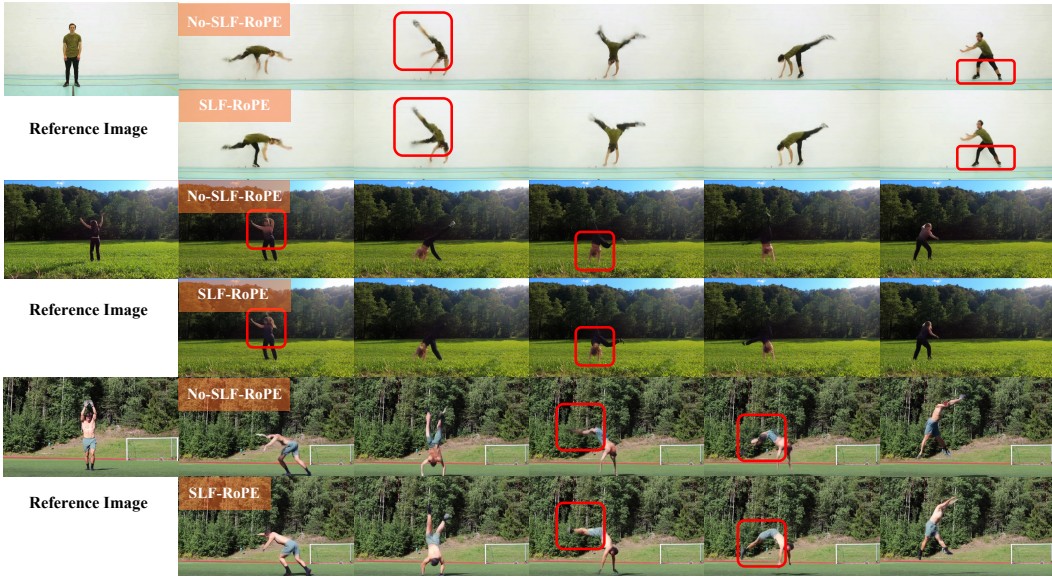

**Figure 8:** Ablation study on the impact of SLF-RoPE on the HyperMotionX Bench. The experimental control group results shown here are all from our models.

The third case further reveals that removal of SLF-RoPE results in severe degradation during airborne motion frames: specifically, the lower limbs of the character become incomplete or distorted, making it difficult to recognize the leg structure. In contrast, the generation results with SLF-RoPE retain the correct appearance and structure, effectively mitigating such artifacts.

**Table 9:** Ablation study on **TikTok** and **UBC-Fashion** benchmarks regarding the impact of SLF-RoPE. ↑ indicates higher is better; ↓ indicates lower is better.

| Model | PSNR ↑ | SSIM ↑ | L1 ↓ | LPIPS ↓ | FID ↓ | FVD ↓ |
|---|---|---|---|---|---|---|
| Remove SLF-RoPE (TikTok) | 16.98 | 0.64 | $89.7 \times 10^{-3}$ | 0.23 | 98.45 | 833.42 |
| SLF-RoPE (TikTok) | 16.52 | 0.63 | $97.4 \times 10^{-3}$ | 0.26 | 102.74 | 887.32 |
| Remove SLF-RoPE (Fashion) | 23.73 | 0.90 | $26.1 \times 10^{-3}$ | 0.052 | 29.80 | 172.47 |
| SLF-RoPE (Fashion) | 23.86 | 0.90 | $26.0 \times 10^{-3}$ | 0.059 | 38.31 | 243.29 |

These qualitative findings align well with the results of our quantitative ablation study, confirming the importance of SLF-RoPE in preserving human appearance and structural integrity in complex motion scenarios.

---

**Algorithm 1** SLF-RoPE Frequency Scaling

---

**Require:** Frequency tensor $\boldsymbol{\theta}$ split into temporal ($t$), height ($h$), and width ($w$) components
**Require:** Learnable parameters motion_scale, space_scale_factor
**Require:** Low-frequency ratio $\alpha = 30\%$
  1: Compute scaling factor $\gamma = 1 + $ space_scale_factor $\cdot$ motion_scale
  2: **for** each spatial axis $a \in \{h, w\}$ **do**
  3:    Identify low-frequency indices: $I_{\text{low}} = $ last $\alpha\%$ of $\boldsymbol{\theta}^{(a)}$
  4:    Scale low-frequency values: $\boldsymbol{\theta}^{(a)}[I_{\text{low}}] \leftarrow \gamma \cdot \boldsymbol{\theta}^{(a)}[I_{\text{low}}]$
  5: **end for**
  6: Apply modified $\boldsymbol{\theta}$ in standard RoPE rotation

---

### A.4 EXAMPLES FROM QUALITATIVE EXPERIMENTAL

In Fig 9 and Fig 10, we compare various state-of-the-art methods. Our approach demonstrates superior pixel fidelity and consistency in limb structure across high-speed rotations, unconventional inverted poses, and somersaults, exhibiting fewer deformations compared to alternative methods. In additional, as shown in Fig 13, our method maintains strong ID consistency and produces accurate hand details even under rapid and large-motion actions.

### A.5 ADDITIONAL TESTING OF INTERNET SOURCE VIDEOS

Fig 11 and Fig 12 demonstrate further the capability of our approach to generate motion transfer animations for target characters using arbitrary video sequences sourced from the internet.

### A.6 LIMITATIONS

While our method demonstrates strong performance in a wide range of complex human motion scenarios, several limitations remain. First, the model still struggles with extremely dynamic and acrobatic movements such as continuous Thomas flares, 540 kicks, and other multi-phase actions. These motions often involve rapid full-body rotations and self-occlusions, which are inherently difficult to reconstruct with high fidelity. Second, under such extreme motions, facial features are often blurred or partially lost, indicating a lack of robustness in preserving fine-grained identity cues during high-speed transitions. Lastly, we observe that even state-of-the-art pose estimators fail to provide accurate hand keypoint sequences in these complex motion segments, making it challenging to guide fine-level motion generation in the hands and fingers.

We believe that future work can address these issues by incorporating explicit face and hand modeling, multiview consistency constraints, or integrating higher-fidelity motion capture priors.

### A.7 USE OF LLMS

The LLMs are used only for language polishing and editing of the manuscript text.

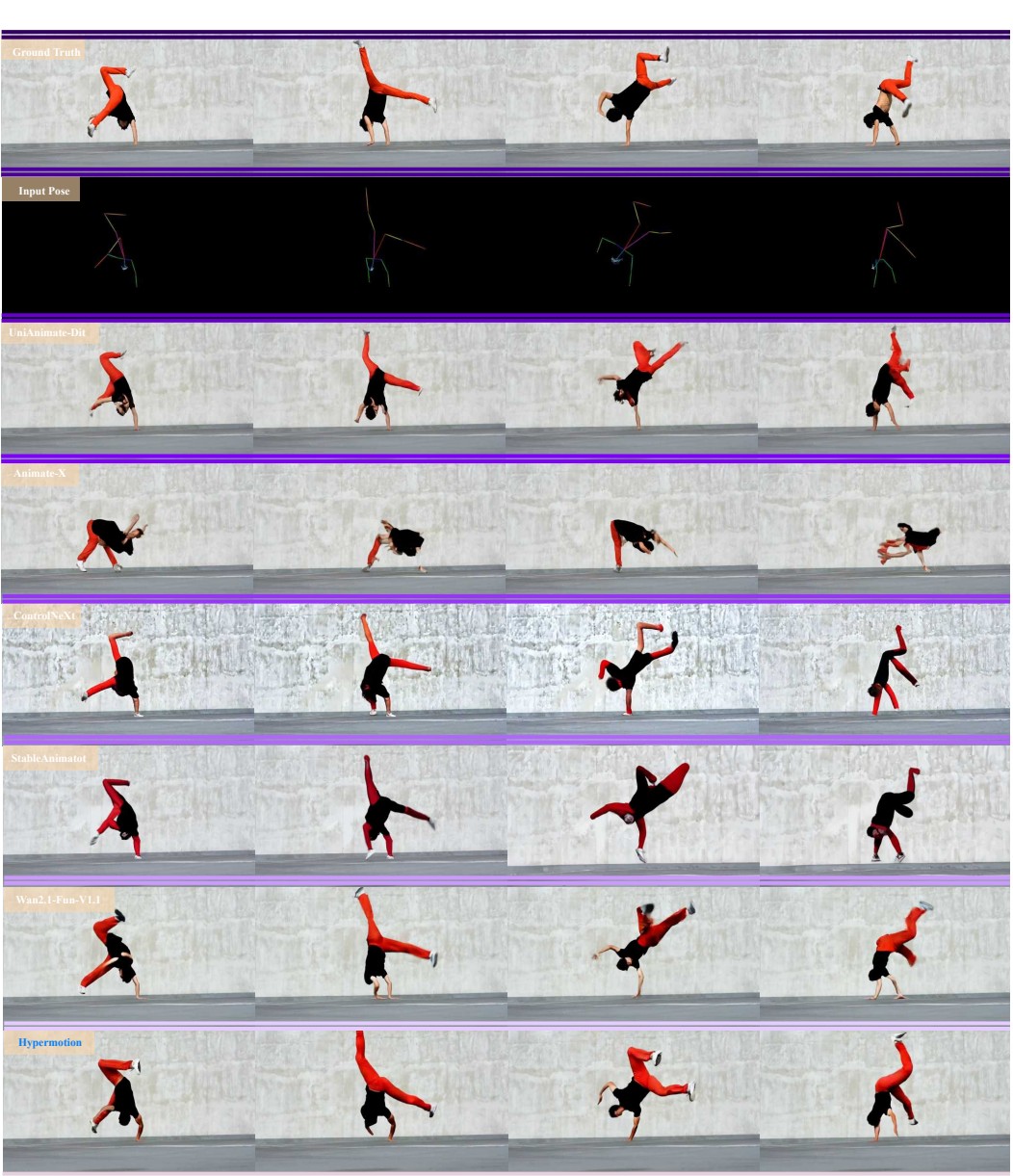

**Figure 9:** Qualitative comparison case *a*.

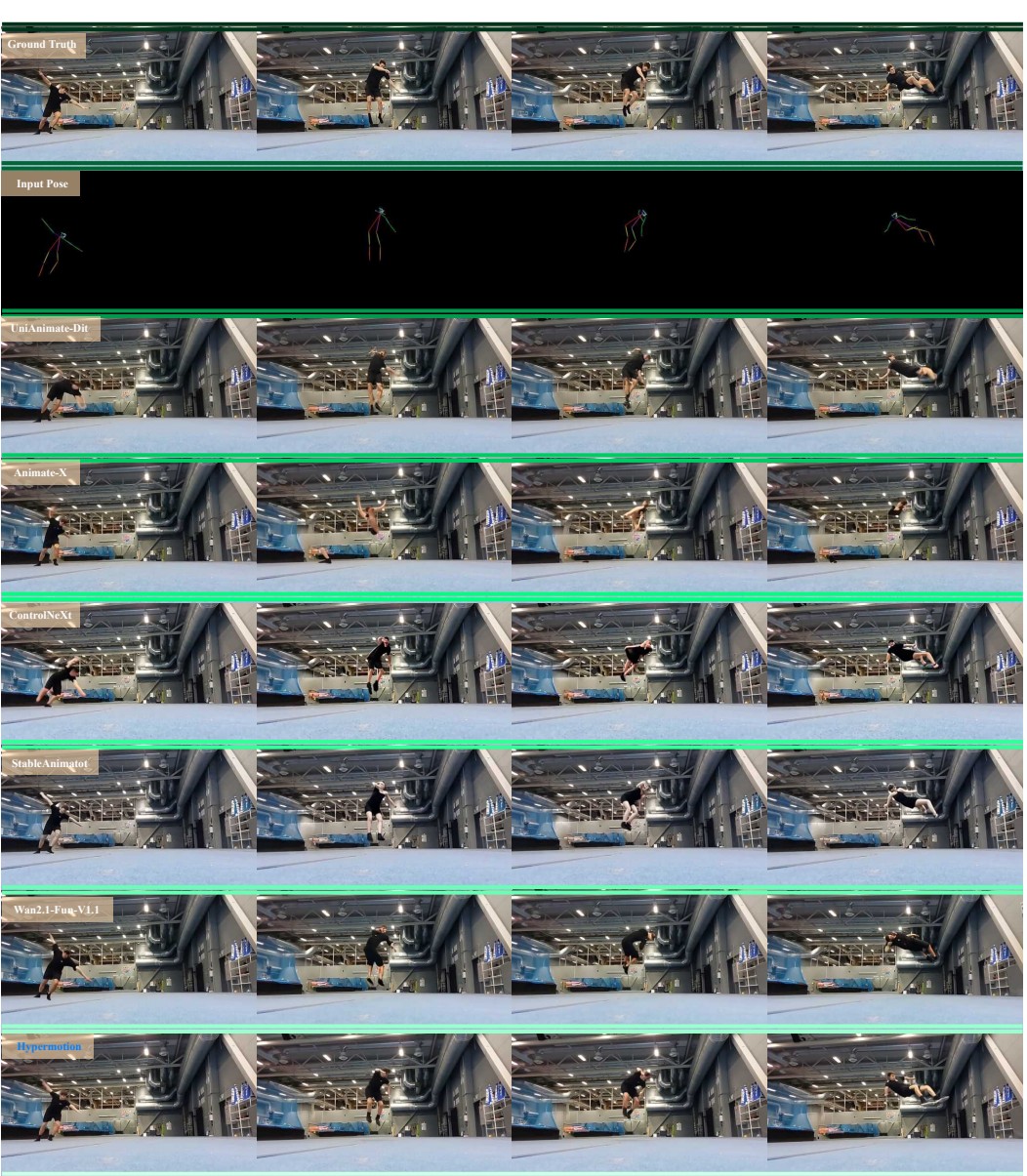

**Figure 10:** Qualitative comparison case *b*.

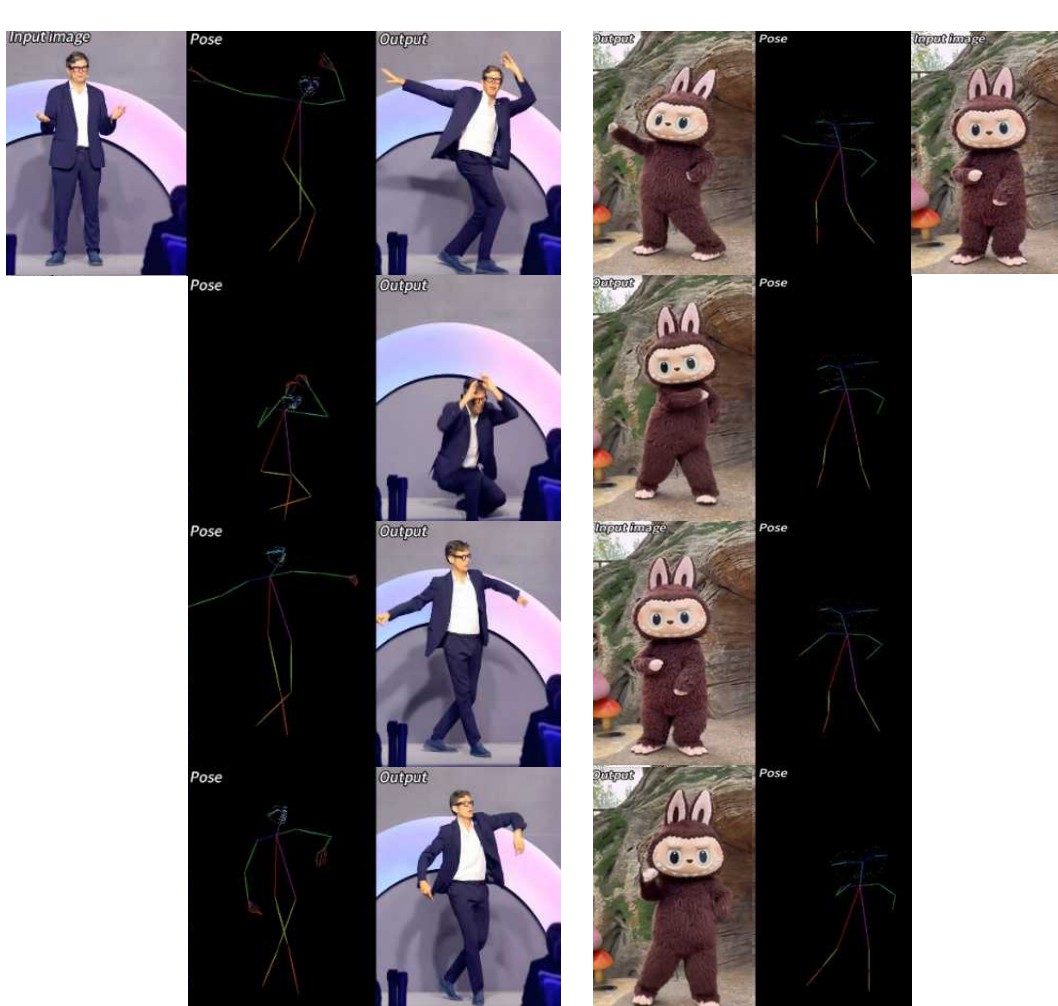

**Figure 11:** Internet source videos testing case *c*.

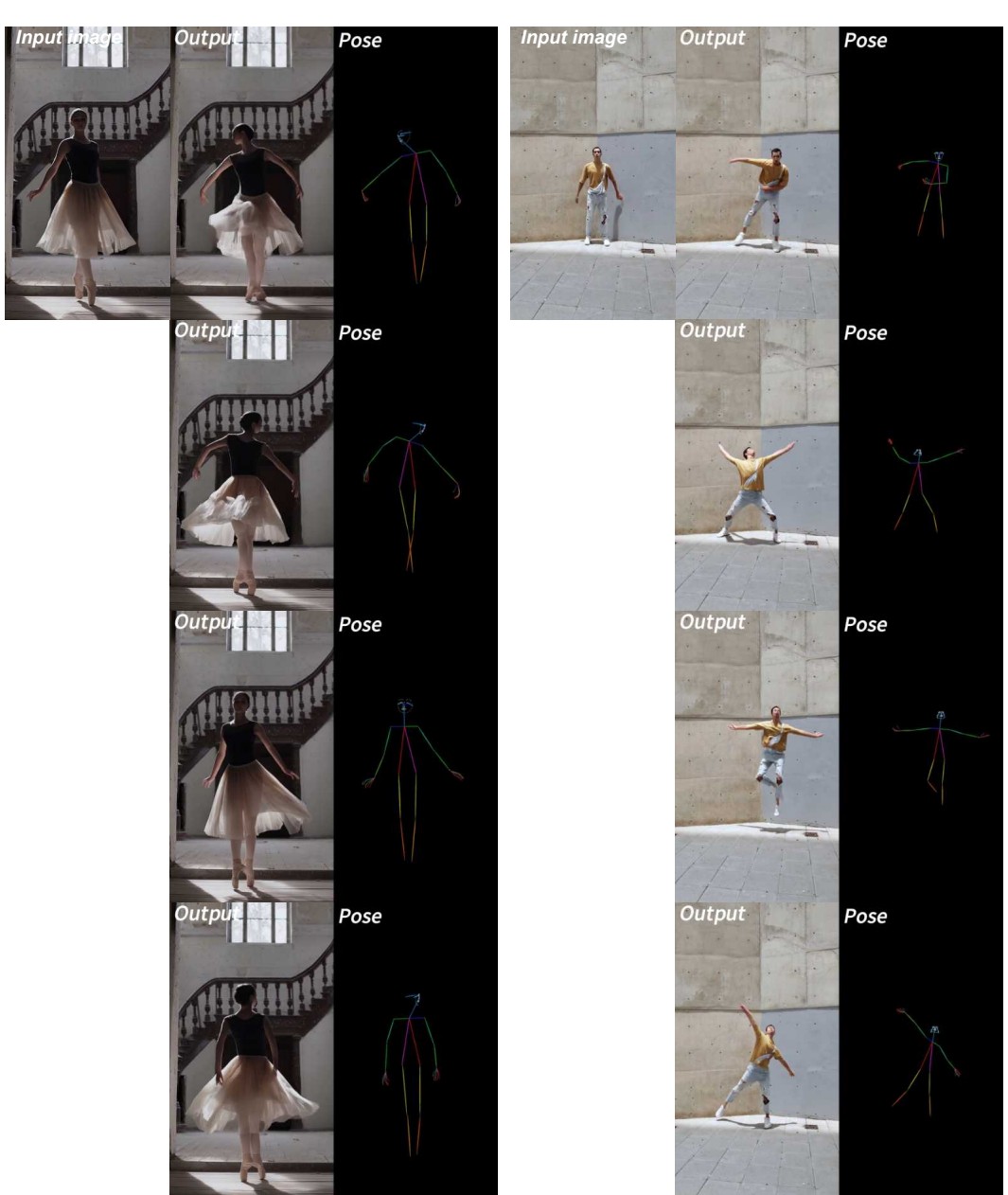

**Figure 12:** Internet source videos testing case *d*.

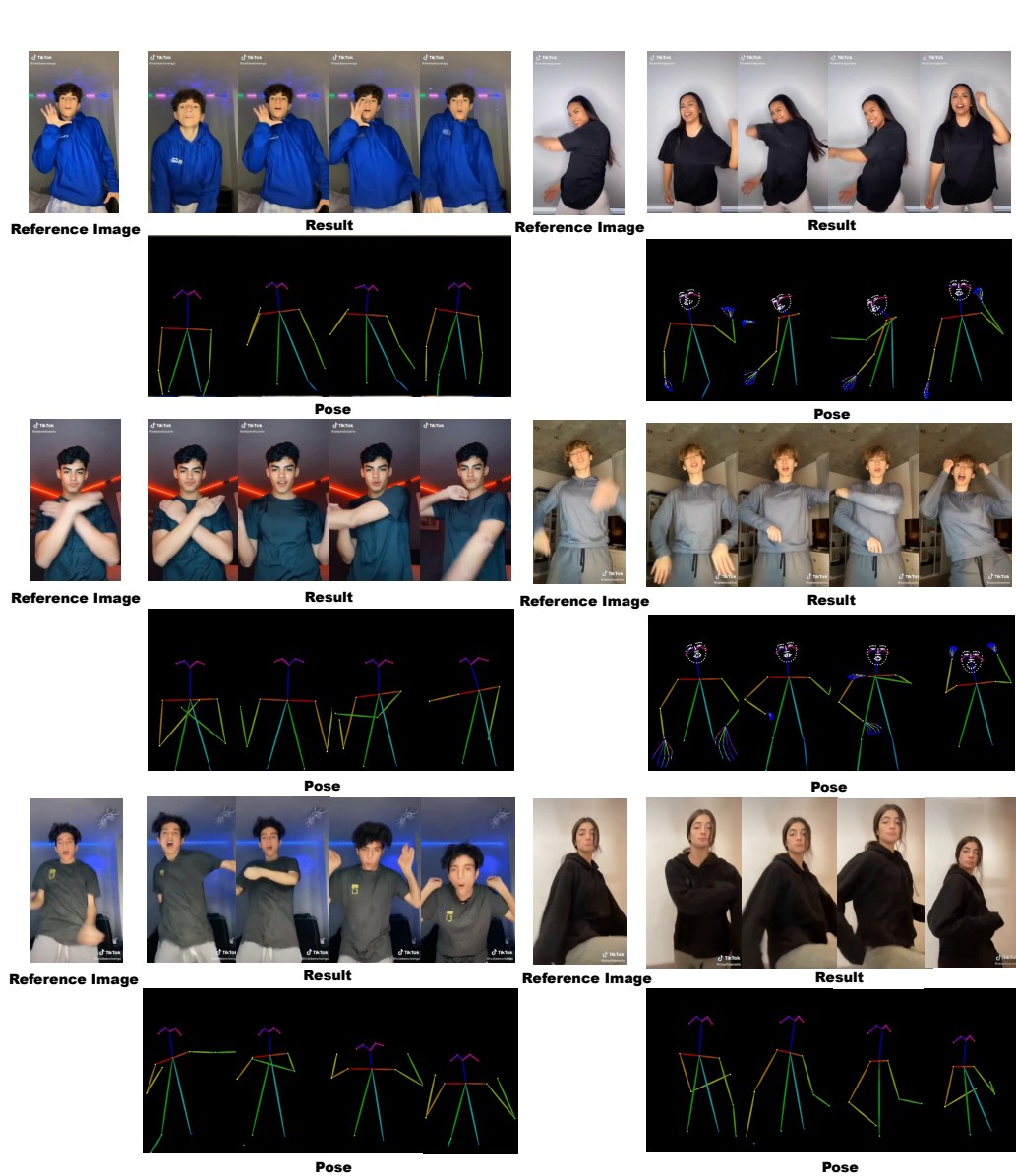

**Figure 13:** Further case studies highlighting reliable hand synthesis and consistent identity preservation in challenging high motion scenes.

