# OpenReview forum: "HyperMotion: DiT-Based Pose-Guided Human Image Animation of Complex Motions"
_ICLR.cc/2026/Conference — Submitted to ICLR 2026_

### Official Review · Reviewer_rE7V · 2025-10-26

**Soundness:** 2
**Presentation:** 3
**Contribution:** 2
**Rating:** 4
**Confidence:** 4

**Summary:**

This paper targets the challenge of pose-guided portrait animation under complex human motions (e.g., flips, stunts) and proposes a concise DiT paradigm: it composes in latent space to inject both pose video and reference image, and uses a first-frame mask to suppress identity leakage. The core modification, SLF-RoPE, selectively enhances the low-frequency channels of RoPE along the spatial dimension, while learnable motion/space scaling factors improve global structure and identity stability under high-speed, nonlinear motion. Based on MotionX, the authors construct the Open-HyperMotionX dataset (automatically mining complex motion segments via wavelet energy, with OCR debiasing and caption cleaning) and release HyperMotionX Bench with 100 high-quality pose annotations (using XPose, removing unreliable hand keypoints in complex frames). Trained on the Wan2.1 backbone, the method achieves leading structural consistency (PCK), competitive pixel/perceptual/temporal metrics, and VBench-I2V gains in background/overall consistency and motion smoothness. Ablations show SLF-RoPE effectively reduces artifacts under extreme poses, and inference runs on a single 3090. Limitations include limited theoretical novelty, insufficient analysis of sensitivity to α/γ and the dynamics of the proposed scaling strategy, coarse-grained ablations, heavy reliance on XPose and hand keypoint quality, and incomplete evaluation for multi-person scenes, strong camera motion, and long sequences. The authors plan to open-source code, data, and evaluation, indicating solid engineering practicality and potential impact.

**Strengths:**

1. Method is simple yet targeted: latent-space composition in a DiT and a first-frame mask suppress identity leakage; SLF-RoPE selectively boosts spatial low-frequency channels and introduces learnable motion/space scaling, significantly improving global structure and identity stability under high-speed, non-linear motion.
2. Significant data and evaluation contributions: constructed Open-HyperMotionX (automatic mining of complex motions, OCR debiasing, and caption cleaning) and released the high-quality HyperMotionX Bench (100 sequences with pose annotations), providing practical training and evaluation resources for complex motion scenarios.
3. High engineering practicality: inference runs on a single 3090, with minimal modifications and easy integration into backbones like Wan2.1; plans to open-source code, data, and evaluation, facilitating community reproduction and extension.

**Weaknesses:**

1. Limited methodological novelty: The core modification (SLF-RoPE) scales the low-frequency band of RoPE’s frequencies. The idea is intuitive and engineering-oriented, but lacks theoretical depth and a justification of generality.

2. I would like to see more visual results for this task, especially hand details (which are one of the core aspects). Please provide more examples focusing on hand motion.

3. The sensitivity of the low-frequency ratio α and the scaling factor γ, as well as the learning dynamics of the motion scale and space scale under different motion intensities, remain unclear. How “motion intensity” is estimated and the consistency between training and inference phases are not elaborated.

4. The paper only provides three comparisons (with/without SLF-RoPE and with/without dataset training). More fine-grained ablations are desirable (e.g., scaling H only or W only, fixing γ without dynamic modulation, using different α partitioning strategies, etc.).

5. There is heavy reliance on XPose for pose annotations; hand keypoints are removed in complex segments. While this is reasonable, it limits the evaluation and optimization of fine-grained hand motion generation.

**Questions:**

See Weaknesses

---

> ### Author Response · Authors · 2025-11-21
> **Response to Reviewer: rE7V**
>
> # Response to Reviewer: rE7V
>
> We appreciate the reviewer’s time and constructive comments and interest in our work.
>
> ## Weakness1: Limited Theoretical Justification of SLF-ROPE
> We thank the reviewer **rE7V** for the thoughtful comment. Our work places a stronger emphasis on constructing the Open-HyperMotionX dataset and benchmark, as well as proposing a simple yet effective baseline for complex human motion generation. Therefore, we have selected Primary Area: datasets and benchmarks. The SLF-RoPE module is introduced as a practical enhancement tailored to this setting. In the revised version, we have added additional analyses and experiments to further validate the effectiveness and generality of SLF-RoPE across multiple settings and elaboration of motivation.
>
> ## Weakness2: More visual results for this task, especially hand details
> We have updated the supplementary material as well as new visual cases and **Fig. 13** of the appendix, which provide clearer and more hand details results. We hope to meet your requirements.
>
> ## Weakness3: Lack of Analysis on Motion Intensity and Scaling Behavior
> Thank you for this thoughtful question. We would like to clarify that our method does not explicitly estimate or rely on a defined notion of “motion intensity” during either training or inference. As shown in Algorithm 1, SLF-RoPE is implemented as a purely learnable frequency scaling mechanism, where the 'motion scale' and 'space scale factor' parameters are optimized end-to-end together with the diffusion model, without requiring any external motion signal or additional supervision.
>
> The low-frequency ratio α and scaling factor γ are therefore standard hyperparameters, not motion-dependent variables. Their effects are consistent in both training and inference, since the same RoPE modification is applied uniformly. Because the module introduces no external cues and no stage-dependent behavior, there is no discrepancy between the two phases.
>
> ## Weakness4: Lack of more refined ablation experiments
> We are most grateful for your meticulous identification of weaknesses. In the updated version of our paper, we have incorporated additional ablation studies within the experimental section, including those concerning enhanced 'H' that different frequency domain enhancement strategies, see **Table 4**. We can observe that increased high-frequency training may even have a detrimental effect on performance. However, owing to time constraints during training, we shall endeavour to supplement these with further ablation experiments in future iterations.
>
> ### Table 4: SLF-ROPE Ablation Study Using Different Frequency Domain Enhancement Strategies Training on HyperMotionX Bench
>
> | Model        | PSNR ↑ | SSIM ↑ | L1 ↓ (×10⁻³) | LPIPS ↓ | FID ↓ | FVD ↓ |
> |--------------|--------|--------|---------------|----------|--------|--------|
> | No Enhance   | 21.78  | 0.69   | 48.3          | 0.1255   | 77.00  | 758.71 |
> | Enhance H    | 21.1395| 0.68   | 52.4          | 0.1409   | 85.68  | 827.27 |
> | Enhance L    | 22.03  | 0.71   | 47.0          | 0.124    | 80.91  | 825.68 |
>
> ## Weakness5: Constraint on Hand Motion Quality Stemming from Pose Annotation Gaps
> Regarding this weakness, hand annotation in large scope, open-world human motion remains a significant challenge in the field. To better support the evaluation of fine grained hand motion, we are currently attempting to use **HaMeR: Hand Mesh Recovery** to obtain additional 3D hand annotations for the benchmark, and then map them back to 2D. This ongoing effort aims to further improve hand-related evaluation quality, to be able to help the community.
>
> ### Reference ###
> [1] Reconstructing Hands in 3D with Transformers. CVPR2024

---

### Official Review · Reviewer_sERN · 2025-10-28

**Soundness:** 2
**Presentation:** 3
**Contribution:** 2
**Rating:** 4
**Confidence:** 4

**Summary:**

This paper addresses the critical challenge of generating high-fidelity, temporally consistent human image animations for complex human motions (e.g., stunts, acrobatics), where existing methods typically fail due to structural degradation and appearance inconsistency. The authors propose a concise yet powerful DiT-based human animation baseline and introduce a novel component: Spatial Low-Frequency Enhanced Rotary Positional Embedding (SLF-ROPE), which selectively amplifies low-frequency spatial features to improve global structure and appearance fidelity. Furthermore, the paper contributes the Open-HyperMotionX Dataset and HyperMotionX Bench, a valuable resource for training and evaluating models specifically on complex human movements. Extensive experiments demonstrate significant improvements over the state-of-the-art.

**Strengths:**

-	This paper addresses the critical challenge of generating high-fidelity, temporally consistent human image animations for complex human motions, and presents SLF-ROPE.
-	Experimental results indicate the effectivenss of the proposed method.

**Weaknesses:**

-	Incremental Architecture: While the DiT-based architecture is effective, it is described as a "simple DiT-based baseline" built upon existing models (Wan2.1). The core novelty is concentrated in the SLF-ROPE module and the data/benchmark. A deeper analysis or discussion on why this specific DiT architecture is superior for complex motions beyond the added SLF-ROPE would strengthen the paper.
-	The Lecun ID in the supplementary material is not maintained well. Could the authors explain the reasons?
-	The paper notes that existing human pose estimation methods often fail on complex motions, motivating the new high-quality dataset. While this solves the problem for training/evaluation, a discussion on how the inference stage would be impacted if only low-quality poses were available from external methods would be beneficial, or a demonstration that the robust SLF-ROPE can better handle noisy pose inputs.

**Questions:**

Can the proposed SLF-ROPE apply to other general video generation tasks?

---

> ### Author Response · Authors · 2025-11-21
> **Response to Reviewer: sERN**
>
> # Response to Reviewer: sERN #
>
> First of all, thank you very much for such detailed and serious review comments, which are very helpful to us!
>
> ## Weakness1: Lack of Discussion on the Advantages of the DiT Architecture ##
> We sincerely appreciate the reviewer’s careful suggestion. In the supplementary material, we have added a detailed analysis explaining why we adopt a DiT-based architecture. According to results reported in widely used benchmarks such as V-Bench, DiT-based open source video generation models particularly Wan2.1, demonstrate superior performance in key aspects such as large motion generation and ID consistency. Motivated by these strengths, we build upon this architecture to better tackle the challenges of complex human motion video synthesis. In addition, our quantitative results also indicate that DiT-based approaches generally exhibit stronger baseline performance compared to U-Net architectures.
>
> ## Weakness2: About Lecun ID in the supplementary material is not maintained well ##
> Regarding this case, we acknowledge that the ID of LeCun in the supplementary material is indeed not ideally preserved, although we still have reasonable confidence in the overall robustness of our model. We believe this degradation may be primarily caused by the occlusion from the glasses and the specular reflections introduced during generation, both of which can negatively impact facial identity preservation. And we have updated the supplementary material and included additional videos (**ID_hand.mp4**) as well as new visual cases in Fig. 13 of the appendix, which provide clearer and more consistent ID-preservation results.
>
> ## Weakness3: Robustness Concerns Under Imperfect Low Quality Pose Inputs ##
> Thank you for pointing this out. We have added an additional analysis on the robustness of the SLF-ROPE module under low quality pose (Dwpose) inputs (see **Table3**), and we have incorporated the corresponding discussion into the ablation section of the main paper. The experiments confirm that SLF-ROPE indeed exhibits strong robustness: even when the provided poses are noisy, degraded, or partially missing, the module is able to leverage the diffusion prior to recover the missing segments and generate plausible and structurally consistent human body.
>
> ### Table 3: SLF-ROPE Ablation Study Using Low Quality Pose (DWpose) on HypermotionX Bench
>
> | Model            | PSNR ↑   | SSIM ↑   | L1 ↓ (×10⁻³) | LPIPS ↓ | FID ↓ | FVD ↓  |
> |------------------|----------|----------|---------------|---------|-------|--------|
> | Remove SLF-R0PE  | 20.6492  | 0.6271   | 60.3          | 0.1523  | 81.04 | 833.58 |
> | SLF-R0PE         | 21.2554  | 0.6749   | 52.1          | 0.1366  | 83.031| 882.51 |
>
>
> ## Question: SLF-ROPE apply to other general video generation tasks. ##
> Thank you for the constructive question. We explored human animation video generation under depth based video control and found that it is indeed feasible. We have uploaded the corresponding results as **Deep_control.mp4** in the updated supplementary materials. In fact, this baseline is generally applicable to video control tasks performed in latent space, and should naturally support a variety of control modalities such as depth, line drawings, trajectories, and similar conditioning signals, however, owing to limited computational resources, we are currently unable to provide verification regarding whether the SLF-ROPE module is beneficial for generating these tasks. Nevertheless, we shall endeavour to conduct experiments within the constraints of available time to enhance this work.

---

### Official Review · Reviewer_QtmM · 2025-10-28

**Soundness:** 3
**Presentation:** 3
**Contribution:** 2
**Rating:** 4
**Confidence:** 5

**Summary:**

This paper introduces a Diffusion Transformer (DiT) based framework for pose-guided human image animation, specifically targeting complex, high-dynamic "Hypermotions." The core novelty lies in the Spatial Low-Frequency Enhanced Rotary Positional Embedding (SLF-ROPE), designed to mitigate structural degradation during extreme movements. The authors also contribute a new dataset and benchmark, Open-HyperMotionX and HyperMotionX Bench. The proposed method shows impressive qualitative results and strong quantitative performance. The idea of explicitly addressing the stability issue in complex movements is valuable. However, significant concerns regarding the fairness of the comparison and the evaluation protocol must be addressed.

**Strengths:**

1.	The paper correctly identifies a major failure mode of current human animation models and discovers their inability to maintain fidelity and consistency during complex, non-standard movements. The goal is clear and impactful.
2.	SLF-ROPE and a new benchmark named Open-HyperMotionX are proposed.
3.	Establishing a new benchmark for complex motion is necessary to drive future research in this domain.

**Weaknesses:**

1.	The primary concern is the potential unfairness of the comparison. Existing state-of-the-art methods are generally trained on large, general-purpose datasets (e.g., videos of daily life, simple movements), which predominantly feature simple motions. The proposed method is specifically trained on the newly introduced HyperMotionX dataset, which focuses on complex motions. If the videos used for evaluation (e.g., the supplementary videos) share a similar distribution or style to the videos in the HyperMotionX training set, the comparison is severely biased. The proposed method is essentially specialized for the test domain, while the baselines are being tested out-of-distribution (OOD) for complex motions.
2.	The paper focuses heavily on complex motions. It is essential to demonstrate that the SLF-ROPE modification does not degrade performance or introduce artifacts when applied to the simple, common motions where existing methods already perform well. A comprehensive evaluation on a standard, general animation benchmark (e.g., TikTok, PATD) is mandatory.
3.	The quality of the supplementary material is inconsistent. The IDs in the supplementary material is not consistently preserved well.

**Questions:**

In Table 3, the proposed method does not achieve best performance in terms of FID, VFID, and FVD. Could the authors explain this phenomenon?

---

> ### Author Response · Authors · 2025-11-21
> **Response to Reviewer: QtmM**
>
> # Response to Reviewer: QtmM #
>
> We appreciate the reviewer’s time and constructive comments and interest in our work.
>
> ## Weakness1: Potential unfairness of the comparison ##
> We thank the reviewer QtmM  for raising this important point. We clarify that the HyperMotionX Benchmark is independently collected from the Internet and does not overlap with the Open-HyperMotionX training set. The benchmark was specifically curated to ensure a distinct data source and distribution, avoiding bias toward our model. Therefore, our method is not evaluated on data that favors its training distribution, and all baselines are tested under the same benchmark conditions. We mentioned in **Section 3.5** hypermotion bench, and will make this distinction clearer in the revised manuscript.
>
> ## Weakness2: Evaluating General Animation bench Robustness about SLF-ROPE ##
> We thank for this valuable suggestion.  Following your recommendation. We further evaluate the impact of SLF-RoPE on general-motion benchmarks, including TikTok and UBC-Fashion. As shown in **Table 2**, the results indicate that SLF-RoPE does not degrade performance on these common-motion datasets, achieving comparable fidelity and perceptual quality to the baseline. This confirms that our frequency-aware modification remains stable on simple motions while primarily benefiting complex-motion scenarios.  We have also been integrated into the revised ablation study section of the main paper.
>
> ### Table 2: Ablation study on **Tiktok, UBC-fashion** Bench sets about SLF-RoPE
>
> | Model                         | PSNR ↑ | SSIM ↑ | L1 ↓ (×10⁻³) | LPIPS ↓ | FID ↓  | FVD ↓  |
> |------------------------------|--------|--------|----------------|----------|--------|--------|
> | Remove SLF-RoPE (Tiktok)     | 16.98  | 0.64   | 89.7           | 0.23     | 98.45  | 833.42 |
> | SLF-RoPE (Tiktok)            | 16.52  | 0.63   | 97.4           | 0.26     | 102.74 | 887.32 |
> | Remove SLF-RoPE (Fashion)    | 23.73  | 0.90   | 26.1           | 0.052    | 29.80  | 172.47 |
> | SLF-RoPE (Fashion)           | 23.86  | 0.90   | 26.0           | 0.059    | 38.31  | 243.29 |
>
>
> ## Weakness3: Inconsistent ID Preservation in Supplementary Material ##
> Thank you for pointing this out. We have updated the supplementary material and included additional videos (ID_hand.mp4) as well as new visual cases in **Fig. 13** of the appendix, which provide clearer and more consistent ID-preservation results. On the other hand, generating high-fidelity IDs under complex, large range, and high speed motions remains a challenging problem, especially when the face occupies only a very small portion of the frame.
>
> ## Question: Explanation for Suboptimal FID/VFID/FVD Performance ##
> We sincerely thank Reviewer QtmM for raising this important question, which helps us further improve our work. We would like to address this phenomenon through a well-established concept in the computer vision community the **fidelity–perceptual trade-off**. In generative modeling, the trade-off between fidelity and perceptual quality is widely recognized. Foundational studies such as “The Perception-Distortion Tradeoff” (Blau & Michaeli, 2018), as well as recent work like “Traversing Distortion-Perception Tradeoff using a Single Score-Based Generative Model” (Wang et al., 2025), have extensively discussed this inherent balance.
>
> As further evidenced by our ablation studies and qualitative results in the supplementary material, the PCKh@0.5 metric clearly shows that our final model achieves superior structural accuracy and pixel level fidelity compared with both the variant without SLF-ROPE and baseline methods. Enhancing structural correctness and fine-grained appearance consistency is precisely the design goal of our approach.
>
> Therefore, this trade-off provides a natural explanation for why our model does not always achieve the best perceptual metrics, while attaining the strongest pixel-level and structural performance.
>
> ### Reference ###
> [1] The Perception-Distortion Tradeoff. (Blau & Michaeli, 2018)
>
> [2]Traversing Distortion-Perception Tradeoff using a Single Score-Based Generative Model. (Wang et al., 2025)

---

> > ### Comment · Reviewer_QtmM · 2025-11-26
> >
> > Thanks to the authors for their replies. Some of my concerns have been addressed. However, I still have the following questions:
> >
> > 1. The authors claim: "We clarify that the HyperMotionX Benchmark is independently collected from the Internet and does not overlap with the Open-HyperMotionX training set." In reality, the distribution of the training data and the distribution of the testing data may overlap to some extent. For example, previous methods are trained on human dancing datasets, which do not involve complex human-object interactions. The training dataset used in this work includes such interactions.
> >
> > 2. The results on general datasets show that the proposed method lacks competitiveness. For example, StableAnimator has a PSNR of 20.66 and FVD of 140.62 on TikTok, results that are far better than the method presented in this paper.
> >
> > 3. Inconsistent ID Preservation in Supplementary Material. Although the authors provided some additional results, this cherry-picking method is not user-friendly and indicates the instability of the proposed method.
> >
> > Therefore, I tend to maintain my score at 4.

---

### Official Review · Reviewer_iPpe · 2025-10-31

**Soundness:** 3
**Presentation:** 3
**Contribution:** 3
**Rating:** 6
**Confidence:** 3

**Summary:**

This paper addresses the challenge of pose-guided human image animation under complex and highly dynamic motions. The authors propose a DiT-based human animation baseline featuring a novel spatial low-frequency enhanced RoPE module that improves low-frequency spatial feature modeling through learnable frequency scaling. To robustly evaluate how well models can perform for highly dynamic motions, the authors curate the Open-HyperMotionX Dataset and HyperMotionX Bench, which provide high-quality human pose annotations for video clips of complex motion. Experiments show that the proposed model achieves better structural stability and appearance consistency in dynamic motion sequences, demonstrating the combined effectiveness of the new dataset and architectural enhancements.

**Strengths:**

- The work fills an important gap by focusing on complex, highly dynamic human motions that existing models struggle to handle.
- The paper clearly identifies the limitations of prior models in capturing low-frequency spatial details and introduces spatial low-frequency enhanced RoPE as a targeted enhancement that is validated through ablation studies.
- The proposed benchmark and dataset are well-curated, clearly documented, and demonstrate advantages in evaluation, providing a valuable resource for future research in pose-guided human animation.

**Weaknesses:**

- The impact of the dataset could be analyzed more extensively. The paper primarily compares HyperMotion vs. Wan's model performance on VBench before and after training on the Open-HyperMotionX, but further evaluation on finetuned versions of other baselines models could strengthen the assessment of benefits for the constructed dataset. If computationally feasible, such comparisons could better isolate the dataset’s contribution from architectural effects.

- The analysis of limitations of current models on clean but hard pose sequences is insightful and motivates the design of low-frequency detail modeling in positional embeddings. However, the performance improvements of the HyperMotion model on the HyperMotionX Benchmark are not consistently significant, even though the model is trained on Open-HyperMotionX, potentially giving it an advantage over other models. This suggests that the model architecture may still have room for improvement.

**Questions:**

- During training, the reference image appears to be randomly selected. Does this imply that during inference, the reference image does not necessarily need to correspond to the first frame of the motion sequence? If so, how sensitive is model performance to this choice of reference image?

- Regarding the wavelet-based clip extraction, how are long dynamic motions, such as dancing or motions with buildup phases (e.g., running before a long jump), handled? Are these sequences split into multiple shorter segments, and if so, does this segmentation risk disrupting motion continuity or long-term temporal patterns during generation?

---

> ### Author Response · Authors · 2025-11-21
> **Response to Reviewer: iPpe**
>
> # Response to Reviewer: iPpe #
>
> First of all, thank you very much for your comments and suggestions on our work!
>
> ## Weakness1: Insufficient Evaluation of Dataset Contributions Across Baselines ##
> We are most grateful for your suggestions regarding enhancing the evaluation of dataset contributions. We have supplemented our experiments with the Open-HyperMotionX dataset for fine-tuning on the EasyAnimation model, which represents a fundamentally distinct baseline designed based on Hunyuan video and MM-dit. Please refer to **Table 1**. Additionally, we have incorporated these experiments into the ablation study section of the main text, demonstrating the beneficial impact of our dataset across different baseline models. Should computational resources permit, we shall conduct further testing with additional baseline models.
>
> ### Table 1: Comparison EasyAnimate5-7B (control-task) Before and After Training used Open-HyperMotionX on HyperMotionX Benchmark
>
> | Model        | PSNR ↑   | SSIM ↑  | L1 ↓ (×10⁻³) | LPIPS ↓ | FID ↓  | FVD ↓    |
> |--------------|----------|---------|---------------|---------|--------|----------|
> | Un-trained   | 19.3069  | 0.6398  | 63.9          | 0.196   | 111.09 | 1377.32  |
> | Trained      | 19.7383  | 0.6551  | 59.4          | 0.188   | 97.44  | 1129.42  |
>
>
>
> ## Weakness2: Limited Performance Gain on HyperMotionX Benchmark ##
> We thank the reviewer **iPpe** for endorsing our analysis of the limitations of current models .  First, we hope clarify that the HyperMotionX Benchmark is independently collected from the internet and does not overlap with the Open-HyperMotionX training set, ensuring a fair evaluation without data leakage or source bias. In additional,  regarding the performance gain, we note that HyperMotionX contains extremely challenging conplex motionwhere improvements tend to be incremental even for recent Sota models. We agree that the model architecture has room for further enhancement, and we consider SLF-ROPE a first step toward more effective motion frequency modeling.
>
> As shown in our results, SLF-RoPE consistently improves pixel-level metrics such as LPIPS, SSIM, and PSNR, which are more indicative of spatial structure and appearance fidelity. These improvements align with our motivation for SLF-RoPE: to enhance intra-frame human structure and pixel-level consistency.
>
> Furthermore, the no-reference metric **VBench-I2** also shows stable improvements, supporting the effectiveness of SLF-RoPE even under high-capacity backbones.
>
> ## Questions1: Reference Image Selection Strategy During Training and Inference ##
> Yes. The reference image is indeed randomly selected during training. This design choice is intentional: by incorporating a masking mechanism, we aim to enable the model to generate results from any arbitrary reference pose, rather than restricting it to the first frame of the motion sequence.
>
> During training, our sampling strategy is as follows:
>
> **1.** At most 40% probability, we use the ground-truth first frame as the reference image.
>
> **2.** For the remaining cases, we randomly sample any other frame from the ground-truth video (excluding the first frame). Combined with the first-frame latent mask, this strategy effectively suppresses identity leakage and enhances the model’s robustness when processing diverse reference inputs.
>
> Overall, our method is designed to support reference images with arbitrary poses. However, similar to most existing approaches, the reference pose and body size should not deviate excessively from the driving pose. In practice, providing a frontal, well-framed reference image generally leads to more stable and higher-fidelity generation results.
>
> ## Questions1: Robustness of Clip Extraction for Long or Build-Up Motions ##
> Thank you for the thoughtful question. Our wavelet-based strategy is designed to preserve complete dynamic motions rather than split them. This is because the method detects instantaneous motion energy through keypoint velocities and then applies CWT-based time frequency analysis to capture how motion intensity evolves over time.
>
> During window selection, we search for the 6 second segment with the highest cumulative wavelet energy. For motions with buildup phasessuch as a run-up before a long jump the run-up and the jump together produce a higher total energy than either part alone. As a result, the algorithm naturally selects the entire coherent motion, including both preparation and peak movement, rather than fragmenting it.
>
> Overall, our method does not disrupt long-term temporal patterns. Instead, it reliably extracts complete composite actions. or videos where entire segments consist solely of dancing, our method can also extract sequences of more vigorous continuous motion. We have applied this approach primarily to the subset of Kung Fu and sports content, as their source videos often contain lengthy segments of low value motion (unrelated to core action content).

---

### Author Response · Authors · 2025-11-26
**Global Reply**

### Dear Reviewers & AC ###

Thank all reviewers for your thoughtful comments and constructive feedback. We have revised the manuscript accordingly. These modifications are primarily contained in Section 4.4, Ablation Study, we have added a total of four sets of various ablation experiments to address the concerns raised by the reviewers.

1. Contributions of Open-HypermotionX evaluated by training on other baseline models.
2.  Ablation experiments for SLF-RoPE on a general common motion benchmark.
3. Assessing the robustness of the SLF-RoPE module using low-quality poses as input.
4. Ablation experiments on different augmentation frequency settings for SLF-RoPE.

### Best regards

### HyperMotion Authors

---

### Meta-Review · Area_Chair_xv7y · 2025-12-28

**Summary:**

**Summary**:
This paper introduces Open-HyperMotionX and HyperMotionX, a new dataset for complex image animation tasks with high-quality human pose annotations, and a benchmark to estimate this task under complex human motion, respectively. To improve the performance on the complext motion animation task, a DiT-based architecture, along with a SLF-ROPE method. The better qualitative and quantitative results on the new benchmark are reported, the fair comparison on other datasets are missed.

**Main strengths**:
- A new dataset and benchmark for the complex image animation task are introduced.
- A SLF-ROPE method is proposed to encode the low-frequency spatial information for the complex motion task.

**Main weaknesses**:
- The impact of the dataset is not fully demonstrated. The results are mainly reported on training and testing on the proposed datasets.
- The proposed method is also mainly accessed on the proposed benchmark. A fair comparison to the SoTA methods on existing datasets is missed.
- Incremental architecture with incremental improvement, and the results reported are worse than the results in the original paper.
- Limited methodological novelty. The proposed architecture is only an incremental modification, and the importance is not well demonstrated.

**Suggested decision**: This paper received initial scores of 6 (iPpe), 4 (QtmM), 4 (sERN), and 4 (rE7V). Only reviewer QtmM replied, but with more concerns. And many other concerns are not fully addressed. Hence, I recommend the final score as "reject".

**Reviewer Concerns:**

**The impact of the dataset needs further analysis (iPpe,QtmM)**: The authors propose additional results by training EasyAnimate5-7B on this dataset, but it is unclear whether this improvement generalizes to other datasets.

**The generalization of the proposed SLF-ROPE (iPpe,QtmM, sERN, rE7V)**: The ablations are provided on Tiktok and Fashion. However, the SoTA baselines show far better results than those reported in this paper (QtmM).

**Unfair comparison (QtmM, rE7V)**: More results are provided.

**Limited methodological novelty (rE7V)**: Not fully addressed. Instead, the authors claimed the novelty of datasets and benchmarks.

**Reviewer Scores:**

The paper initially received scores of 6 (iPpe), 4 (QtmM), 4 (sERN), and 4 (rE7V). Only reviewer QtmM replied, "Some of my concerns have been addressed. I tend to maintain my score at 4." And the authors did not further address the reviewer's comments. As a result, these reviewers are likely to retain the original scores.

---

### Decision · Program_Chairs · 2026-01-26

Reject